# Debiasing Synthetic Data Generated by Deep Generative Models

**Alexander Decruyenaere** *
Ghent University Hospital – SYNDARA

**Heidelinde Dehaene** *
Ghent University Hospital – SYNDARA

**Paloma Rabaey**
Ghent University – imec

**Christiaan Polet**
Ghent University Hospital – SYNDARA

**Johan Decruyenaere**
Ghent University Hospital – SYNDARA

**Thomas Demeester**
Ghent University – imec

**Stijn Vansteelandt**
Ghent University – Department of Applied Mathematics,
Computer Science and Statistics

*\*Joint first authors and corresponding authors*
`{firstname.lastname}@ugent.be`

## Abstract

While synthetic data hold great promise for privacy protection, their statistical analysis poses significant challenges that necessitate innovative solutions. The use of deep generative models (DGMs) for synthetic data generation is known to induce considerable bias and imprecision into synthetic data analyses, compromising their inferential utility as opposed to original data analyses. This bias and uncertainty can be substantial enough to impede statistical convergence rates, even in seemingly straightforward analyses like mean calculation. The standard errors of such estimators then exhibit slower shrinkage with sample size than the typical 1 over root-$n$ rate. This complicates fundamental calculations like p-values and confidence intervals, with no straightforward remedy currently available. In response to these challenges, we propose a new strategy that targets synthetic data created by DGMs for specific data analyses. Drawing insights from debiased and targeted machine learning, our approach accounts for biases, enhances convergence rates, and facilitates the calculation of estimators with easily approximated large sample variances. We exemplify our proposal through a simulation study on toy data and two case studies on real-world data, highlighting the importance of tailoring DGMs for targeted data analysis. This debiasing strategy contributes to advancing the reliability and applicability of synthetic data in statistical inference.

## 1 Introduction

The concept of generating synthetic data as a means of privacy protection was initially introduced by Rubin (1993) within the framework of multiple imputation, a widely used technique for managing the statistical analysis of incomplete data sets. Since its inception, a substantial body of literature on synthetic data has emerged (Raghunathan et al., 2003; Raghunathan, 2021; Drechsler, 2011; Raab et al., 2016; Reiter, 2005), with a recent surge in interest propelled by advancements in deep generative modelling technology (Raghunathan, 2021; van Breugel et al., 2023; Wan et al., 2017; Yan et al., 2022; Nowok et al., 2016; Endres et al., 2022; Hernandez et al., 2022). In this work, we focus on tabular synthetic data and their inferential utility, which captures whether synthetic data can be used to obtain valid estimates and inference for a population parameter (Decruyenaere et al., 2024).

38th Conference on Neural Information Processing Systems (NeurIPS 2024).

The substantial privacy protection potential offered by synthetic data is marred by significant challenges that undermine their inferential utility (Raab et al., 2016). Previous work by Decruyenaere et al. (2024) has shown that these challenges are much more severe when using deep generative models (DGMs), rather than parametric statistical models, which is why we focus on the former. First, standard confidence intervals and p-values obtained on synthetic data may drastically underestimate the uncertainty in synthetic data as these ignore the size of the original data. Indeed, synthetic data obtained via generators trained on small datasets will unsurprisingly deliver much worse quality than synthetic data obtained via generators trained on large datasets. This uncertainty in the generator must therefore be translated into analysis results, such as confidence intervals and p-values. Standard confidence intervals and p-values ignore this, as they do not distinguish whether the data are synthetic or real. Second, it is well known from the literature on plug-in estimation that data-adaptive methods (such as DGMs) cannot succeed to estimate *all* features of the data-generating distribution well (Bickel et al., 1993; van der Laan and Rose, 2011; Chernozhukov et al., 2018; Hines et al., 2022). These methods are designed to optimally balance bias and variance w.r.t. a chosen criterion, such as prediction error. However, they cannot guarantee that such an optimal trade-off is simultaneously made w.r.t. all possible discrepancy measures that exist, such as mean squared error in specific functionals (mean, variance, least squares projections...) of the observed data distribution. Such data-adaptive methods therefore leave non-negligible bias in estimators of such functionals, leading to excess variability, slow convergence, and confidence intervals that do not cover the truth at nominal level (and may even never contain the truth, even in large samples) (Decruyenaere et al., 2024).

**Related work.**   While several approaches have been proposed to account for the uncertainty arising from synthetic data generation, we are not aware of strategies for generating and analysing DGM-based synthetic data that guarantee valid inference. Raghunathan et al. (2003) developed a framework inspired by the work on multiple imputation for missing data, by combining the results of multiple synthetic datasets, but this is not readily applicable to DGM-based synthetic data. Räisä et al. (2023) extended this work for differentially private (DP) synthetic data, acknowledging the additional DP noise, but continue to consider parametric (Bayesian) data generation strategies.

Our work instead focuses on obtaining valid inference from a single synthetic dataset, which is arguably more attractive for use by practitioners. Raab et al. (2016) derived alternative combining rules that reduce to the correction factor $\sqrt{1 + m/n}$ for the standard error (SE) of an estimator in the case of inference from a single (non-DP) synthetic dataset of size $m$ generated from an original dataset of size $n$. The method suggested by Awan and Cai (2020) to preserve efficient estimators in a single (DP or non-DP) synthetic dataset relies on generating data conditional on the estimate in the original data. Both procedures are only applicable to parametric generative models and therefore suffer from the same limitation as the aforementioned approaches. To enable Bayesian inference from a single DP synthetic dataset, Wilde et al. (2021) proposed a corrected analysis that relies on the availability of additional public data, while Ghalebikesabi et al. (2022) investigated importance weighting methods to remove the noise-related bias, but they do not study the impact on inference.

**Contributions.**   Our work is the first to propose a generator-agnostic solution that mitigates the impact of the typical slower-than-$\sqrt{n}$-convergence of estimators in synthetic data created by DGMs. As far as we are aware, our approach is thus the only one that provides some formal guarantees for (more) honest inference in this setting. In this paper, we show how the statistical bias in estimators can be removed, by adapting results on debiased or targeted machine learning (van der Laan and Rose, 2011; Chernozhukov et al., 2018) to the current setting where machine learning is not necessarily used in the analysis, but rather in the generation of synthetic data. Although we build upon ideas from existing work, our extension is non-trivial, since (1) previous work did not consider synthetic data; and (2) we demonstrate, with significant generality, how to mitigate the estimation errors in the DGM that would otherwise propagate into the estimator calculated on synthetic data.

In Section 2 and 3, we propose a generator-agnostic debiasing strategy, directed towards the downstream statistical analysis of the synthetic data. As such, we obtain estimators that are less sensitive to the typical slow (statistical) convergence affecting the generators. We illustrate this with a simulation study in Section 4, showing that the coverage of both the mean and linear regression coefficient estimators indeed improves. In Section 5, we further cement the utility of our debiasing strategy in a practical setting through two case studies. While the proposed strategy is generator-agnostic, we focus our analyses on two DGMs for tabular data: CTGAN and TVAE (Xu et al., 2019). Finally, Section 6 concludes with a discussion on our method and its limitations.

## 2 Notation and Set-Up

The aim of this paper is to use synthetic data in order to learn a specific functional $\theta(.)$ of the observed data distribution $P$. We formalise the problem of learning $\theta(P)$ based on synthetic data as follows. Suppose that, based on $n$ independent (possibly high-dimensional) samples $O_1, ..., O_n$ from $P$, we estimate the observed data distribution as $\widehat{P}_n$. Here, $\widehat{P}_n$ may be obtained by fitting parametric models to the distribution of the observed data, or alternatively by training DGMs. Based on $m$ independent (synthetic) random samples $S_1, ..., S_m$ from $\widehat{P}_n$, we then estimate $\theta(P)$. For this, the data analyst (to whom $\widehat{P}_n$ is unknown) uses the $m$ synthetic samples $S_1, ..., S_m$ to approximate $\widehat{P}_n$ as $\widetilde{P}_m$, and obtains an estimator of $\theta(P)$ given by $\theta(\widetilde{P}_m)$. We use $\mathbb{P}_n$ to denote the empirical distribution of the observed data and $\widetilde{\mathbb{P}}_m$ to denote the empirical distribution of the synthetic data. We index expectations by the distribution under which they are taken; e.g., $E_P(Y)$ denotes the population expectation of $Y$.

Throughout, we assume that the parameter of interest is sufficiently smooth in the data-generating distribution so that root-$n$ consistent estimators (i.e., with SEs shrinking at 1 over root-$n$ rate) can be obtained. Formally, we assume the $\theta(P)$ is pathwise differentiable with efficient influence curve (EIC) $\phi(., P)$ under the nonparametric model, as is satisfied for (most) standard statistical analyses. The EIC is a functional derivative of $\theta(P)$ w.r.t. the data-generating distribution $P$ (in the sense of a Gateaux derivative), which has mean zero under $P$ (Fisher and Kennedy, 2021; Hines et al., 2022).

In what follows, we illustrate our debiasing strategy via two examples. Here, we briefly outline some of their theoretical foundations we build upon. Appendix A.1 clarifies how these debiased estimators originate from their EIC. We further refer to them as debiased or EIC-based estimators.

**Population mean.** Adapting the traditional formulation of the population mean to the context of synthetic data, we choose $\widetilde{P}_m = \widetilde{\mathbb{P}}_m$. As a result, we will study the large sample behaviour of the synthetic data sample average:

$$\theta(\widetilde{P}_m) = \frac{1}{m} \sum_{i=1}^{m} S_i.$$

The EIC of this sample average is $\phi(S, \widehat{P}_n) = S - \theta(\widehat{P}_n)$.

**Linear regression coefficient.** Second, suppose we are interested in a specific coefficient $\theta \equiv \theta(P)$ of some exposure $A$ for the outcome $Y$ in the linear model $E_P(Y|A, X) = \theta A + \omega(X)$, where $X$ is a possibly high-dimensional vector of covariates and $\omega(.)$ is an unknown function. Our proposal allows $\omega(X)$ to correspond to a standard linear model in all covariates, but is more generally valid. Building upon the nonparametric definition of $\theta$ as given in Appendix A.1, we adjust it to obtain an estimator in the setting of synthetic data. Let $Y$, $X$ and $A$ be jointly or sequentially modelled by some generative model, from which a single synthetic dataset $S_i = (\widetilde{Y}_i, \widetilde{A}_i, \widetilde{X}_i)$ is sampled ($i = 1, \ldots, m$). We then consider the estimated regression coefficient of exposure $A$ given by

$$\theta(\widetilde{P}_m) = \frac{\frac{1}{m} \sum_{i=1}^{m} \left\{ \widetilde{A}_i - E_{\widetilde{P}_m}(A|\widetilde{X}_i) \right\} \left\{ \widetilde{Y}_i - E_{\widetilde{P}_m}(Y|\widetilde{X}_i) \right\}}{\frac{1}{m} \sum_{i=1}^{m} \left\{ \widetilde{A}_i - E_{\widetilde{P}_m}(A|\widetilde{X}_i) \right\}^2},$$

where the nuisance parameters $E_{\widetilde{P}_m}(A|\widetilde{X}_i)$ and $E_{\widetilde{P}_m}(Y|\widetilde{X}_i)$ are estimated based on synthetic data. This EIC-based estimator coincides with the Maximum Likelihood Estimator (MLE) when least squares predictions for these nuisance parameters are used. Its EIC is (Vansteelandt and Dukes, 2022)

$$\phi(S, \widehat{P}_n) = \frac{\left\{ \widetilde{A} - E_{\widehat{P}_n}(A|\widetilde{X}) \right\} \left[ Y - E_{\widehat{P}_n}(Y|\widetilde{X}) - \theta(\widehat{P}_n) \left\{ \widetilde{A} - E_{\widehat{P}_n}(A|\widetilde{X}) \right\} \right]}{E_{\widehat{P}_n} \left[ \left\{ \widetilde{A} - E_{\widehat{P}_n}(A|\widetilde{X}) \right\}^2 \right]}.$$

## 3 Methodology

In a first step, we establish an understanding of how much the estimate based on synthetic data $\theta(\widetilde{P}_m)$ differs from the population parameter of interest $\theta(P)$. For this, we study the difference $\theta(\widetilde{P}_m) - \theta(P)$, for which we consider 2 von Mises expansions (i.e., functional Taylor expansions).

In Appendix A.2 we derive that

$$
\begin{aligned}
\theta(\widetilde{P}_m) - \theta(P) \;=\;& \frac{1}{m}\sum_{i=1}^{m}\phi(S_i, \widehat{P}_n) - \frac{1}{m}\sum_{i=1}^{m}\phi(S_i, \widetilde{P}_m) + R(\widetilde{P}_m, \widehat{P}_n) \\
& + \int \left\{ \phi(S, \widetilde{P}_m) - \phi(S, \widehat{P}_n) \right\} d(\widetilde{\mathbb{P}}_m - \widehat{P}_n) \quad\quad (1) \\
& + \frac{1}{n}\sum_{i=1}^{n}\phi(O_i, P) - \frac{1}{n}\sum_{i=1}^{n}\phi(O_i, \widehat{P}_n) + R(\widehat{P}_n, P) \\
& + \int \left\{ \phi(O, \widehat{P}_n) - \phi(O, P) \right\} d(\mathbb{P}_n - P). \quad\quad (2)
\end{aligned}
$$

We now examine the eight terms of this equation and discuss why some may be negligible while other may introduce bias. First, $R(\cdot)$ are remainder terms, which can generally be shown to be small, but must be studied on a case-by-case basis; see Hines et al. (2022) for worked out examples. In order for these remainder terms to be $o_p(m^{-1/2})$ and $o_p(n^{-1/2})$, we generally need that faster than $n^{-1/4}$ convergence rates are obtained for the unknown functionals of $\widetilde{P}_m$ and $\widehat{P}_n$ that appear in the EICs. It is unknown whether these convergence rates are attainable for DGMs; whether they are, will partly depend on the number of parameters in the DGM itself, the dimension of the data and the complexity of the observed data distribution. The simulation study in Section 4 will give further insight into this.

Second, the two empirical process terms (1) and (2) in the von Mises expansion can be shown to be $o_p(m^{-1/2})$ and $o_p(n^{-1/2})$ by Markov's inequality, under weak conditions. For (1) to converge to zero, we will need the difference between $\widetilde{P}_m$ and $\widehat{P}_n$ to converge to zero (in $L_2(\widehat{P}_n)$ at any rate), and the estimator to be calculated on a different part of the data than the one on which $\widetilde{P}_m$ was estimated (Chernozhukov et al., 2018). For (2) to converge to zero, we will need $\widehat{P}_n$ (e.g., the DGM) to consistently estimate $P$ (in $L_2(P)$ at any rate). In addition, it can be argued that the DGM needs to be trained on a different part of the data than that on which the debiasing step (see later) will be applied. In Appendix A.3, we elaborate on the necessity of sample splitting.

We thus foresee that the two remainder and two empirical process terms are negligible under certain conditions. Third, as elaborated in Appendix A.2, we show that the large sample behaviour of both

$$
\frac{1}{n}\sum_{i=1}^{n}\phi(O_i, P) \quad\quad (3) \quad\quad \text{and} \quad\quad \frac{1}{m}\sum_{i=1}^{m}\phi(S_i, \widehat{P}_n) \quad\quad (4)
$$

is standard, and well understood; in particular, these terms vary around zero with variance that can be estimated well. By contrast, the terms

$$
-\frac{1}{m}\sum_{i=1}^{m}\phi(S_i, \widetilde{P}_m) \quad\quad (5) \quad\quad \text{and} \quad\quad -\frac{1}{n}\sum_{i=1}^{n}\phi(O_i, \widehat{P}_n) \quad\quad (6)
$$

need some further discussion. Term (5) generally fails to have mean zero because the synthetic data $S_i$ do not originate from the distribution $\widetilde{P}_m$. This term therefore induces a bias in $\theta(\widetilde{P}_m)$ that results from using data-adaptive estimates $\widetilde{P}_m$ on the synthetic data; a similar term would appear if we instead analysed the real data. Term (6) likewise fails to have mean zero because the observed data $O_i$ do not originate from the distribution $\widehat{P}_n$. Also this term thus induces a bias, now resulting from the use of a generative model to obtain $\widehat{P}_n$. It may be large relative to (3) when DGMs are used, because of slow convergence of $\widehat{P}_n$. It is precisely this term that causes estimators based on synthetic data to converge slowly with increasing sample size, as observed in Decruyenaere et al. (2024).

After identifying the two problematic terms, we now propose in a second step a targeting or debiasing strategy to remove these bias terms (5) and (6). As in van der Laan and Rose (2011) and Chernozhukov et al. (2018), we will remove bias term (5) by analysing the data with debiased estimators based on the EIC that ensure that this bias term then becomes zero. Novel to our proposal is that we will additionally shift the generated data to ensure that also bias term (6) becomes zero. This bias term depends on the EIC, which itself depends on the target parameter of interest. In the next two paragraphs, we discuss how this can be done for the two considered estimators. Note that the proposed strategy does not require any actual finetuning or retraining of the DGM. For a given parameter of interest, a mere post-processing of the synthetic samples, based on access to the DGM as well as

the original data it was trained on, allows eliminating the bias for a given parameter of interest. A graphical summary of the problem setting and our debiasing strategy can be found in Appendix A.4.

## 3.1 Population mean

For the population mean, the debiasing step with respect to term (5) is implicit since the traditional estimator as given in Section 2 is a debiased estimator. Therefore, the proposal to debias a given DGM with respect to the population mean $\theta(P) = \int o\, dP(o)$ amounts to first training the DGM and then augmenting the output for the variable of interest to ensure that bias term (6) is zero, or hence

$$\frac{1}{n}\sum_{i=1}^{n}\phi(O_i,\widehat{P}_n) = \frac{1}{n}\sum_{i=1}^{n}O_i - \theta(\widehat{P}_n) = \overline{O} - \theta(\widehat{P}_n) = 0.$$

In other words, the population mean of the synthetic data under the DGM should match the sample average of the real data. Here, $\theta(\widehat{P}_n)$ can be approximated by generating a very large sample $k$ (e.g., one million samples) based on the DGM and calculating their sample mean $\overline{Y}$. The generative model must then be corrected, to ensure that this sample mean equals $\overline{O}$. To obtain a debiased synthetic sample, we shift all samples generated by the given DGM by adding $\overline{O} - \overline{Y}$ to the considered variable. Note that the sample average of such a set of $m$ corrected synthetic samples will generally differ from $\overline{O}$.

## 3.2 Linear regression coefficient

For linear regression coefficients, this section shows how the samples generated by a given DGM need to be adapted to eliminate bias term (6), written as follows (see Appendix A.5):

$$b = \frac{\sum_{i=1}^{n}\left\{A_i - E_{\widehat{P}_n}(A|X_i)\right\}\left\{Y_i - E_{\widehat{P}_n}(Y|X_i)\right\}}{\sum_{i=1}^{n}\left\{A_i - E_{\widehat{P}_n}(A|X_i)\right\}^2} - \theta(\widehat{P}_n). \tag{7}$$

To compute this bias, we must generate, for each observed value $X_i$, a very large sample of measurements of $A$ and $Y$ with the given level $X_i$, based on the DGM. Then $E_{\widehat{P}_n}(Y|X_i)$ and $E_{\widehat{P}_n}(A|X_i)$ can be estimated as the sample average of those values for respectively $Y$ and $A$. Further based on a very large DGM-generated sample $k$ (e.g. one million samples), we calculate $\theta(\widehat{P}_n)$ as follows:

$$\frac{\sum_{j=1}^{k}\left\{A_j - E_{\widehat{P}_n}(A|X_j)\right\}\left\{Y_j - E_{\widehat{P}_n}(Y|X_j)\right\}}{\sum_{j=1}^{k}\left\{A_j - E_{\widehat{P}_n}(A|X_j)\right\}^2}. \tag{8}$$

Debiasing of the DGM can now be done by adding the product $b\left\{\widetilde{A}_i - E_{\widehat{P}_n}(A|\widetilde{X}_i)\right\}$ to the synthetic outcome observations $\widetilde{Y}_i$ generated by the DGM. An in-depth elaboration is provided in Appendix A.5. With the proposed shifting, we ensure that the debiasing with respect to term (6) is completed. We then proceed our analysis with these shifted synthetic observations and employ the EIC-based estimator of Section 2, which ensures that bias term (5) equals zero as well.

## 3.3 Properties

To summarise, when the remainder terms $R(.)$ and the empirical process terms (1) and (2) are all $o_p(m^{-1/2})$ and $o_p(n^{-1/2})$ and, furthermore, the suggested debiasing approach is used so as to remove terms (5) and (6), then we expect that

$$\theta(\widetilde{P}_m) - \theta(P) \quad = \quad \frac{1}{m}\sum_{i=1}^{m}\phi(S_i,\widehat{P}_n) + \frac{1}{n}\sum_{i=1}^{n}\phi(O_i,P) + o_p(n^{-1/2}) + o_p(m^{-1/2}).$$

In particular, the resulting estimator $\theta(\widetilde{P}_m)$ may even converge at root-$n$ rates (under standard conditions of pathwise differentiability (Bickel et al., 1993; Hines et al., 2022)) and has an easy-to-calculate variance that acknowledges the uncertainty in the generation of synthetic data (provided that the statistical convergence of the generator is not too slow). In Appendix A.6, we show that the variance of $\theta(\widetilde{P}_m)$ may be approximated by

$$\left(\frac{1}{m} + \frac{1}{n}\right)E\left\{\phi(O,P)^2\right\},$$

thereby generalising results known for parametric synthetic data generators (Raab et al., 2016; Decruyenaere et al., 2024), where the correction factor $\sqrt{1 + m/n}$ was proposed. Thus, when $m \to \infty$, the debiased estimator $\theta(\widetilde{P}_m)$ based on debiased synthetic data has the same distribution as when the real data were analysed. Therefore, the proposed debiasing strategy delivers analysis results that are asymptotically equivalent to those obtained from the same analysis on the real data, provided that $n/m = o(1)$. This means that results of the same quality and confidence intervals of the same expected length are then obtained, as will be illustrated in the case study in Section 5.2.

Since $E\left\{\phi(O, P)^2\right\}$ is unknown, it merely remains to estimate it as

$$\frac{1}{m} \sum_{i=1}^{m} \phi(S_i, \widetilde{P}_m)^2.$$

This is a consistent estimator when $\widetilde{P}_m$ converges to $\widehat{P}_n$ as $m$ goes to infinity, and moreover, $\widehat{P}_n$ converges to $P$ as $n$ goes to infinity. We note that this sample variance will be subject to bias that results from 'poor' tuning of the DGM. Removing this bias is not required, because this variance will be scaled by $1/m + 1/n$ so that any bias becomes negligible in large samples. While the use of debiased estimators based on the EIC of $E\left\{\phi(O, P)^2\right\}$ may potentially improve performance, this goes beyond the scope of this work.

**Practical implications.** Sections 3.1 and 3.2 show how bias term (6) is eliminated by shifting the synthetic variable of interest. We describe how bias term (5) is also removed by using debiased estimators, which we referred to as EIC-based estimators (see Section 2). As mentioned earlier, the EIC-based estimator for the population mean always coincides with the sample average, while for the linear regression coefficient it only reduces to the ordinary least squares estimator when data-adaptive estimation of the nuisance parameters is not used. This implies that it may suffice for the applied researcher to 1) shift the synthetic data and apply the traditional estimators, and 2) to multiply the SE of the estimator with $\sqrt{1 + m/n}$ to obtain valid inference from a single synthetic dataset. However, the EIC-based estimators are recommended since they are robust against model misspecification by allowing for more flexibility in the estimation of the nuisance parameters.

# 4 Simulation study

Our proposed debiasing strategy is empirically validated by a simulation study that covers both estimators 3.1 (sample mean) and 3.2 (linear regression coefficient). Having full control over the data generating process allows us to calculate the bias, SE and convergence rate of both estimators in synthetic data, with and without our debiasing strategy. The data generating process consists of the following four variables: *age* (normally distributed), atherosclerosis *stage* (ordinal with four categories), *therapy* (binary), and *blood pressure* (normally distributed). The Directed Acyclic Graph (DAG) in Figure 1 represents the dependency structure and we refer to Appendix A.7.1 for more details. This setting allows us to simultaneously target the population mean of *age* and the population effect of *therapy* ($A$) on *blood pressure* ($Y$) adjusted for *stage* ($X$).

$$\text{age} \longrightarrow \begin{array}{c}\text{atherosclerosis}\\\text{stage}\end{array} \longrightarrow \begin{array}{c}\text{blood}\\\text{pressure}\end{array} \longleftarrow \text{therapy}$$

Figure 1: DAG for the variables in the simulation study.

## 4.1 Set-up

We conduct a Monte Carlo simulation study, where $n$ independent records are sampled from the data generating process, forming the observed **original dataset** $O_1, ..., O_n$. This process is repeated 250 times, with the sample size $n$ varying log-uniformly between 50 and 5000 (i.e., $n \in \{50, 160, 500, 1600, 5000\}$). Per original dataset, following DGMs are trained: `CTGAN` and `TVAE` (Xu et al., 2019), of which a detailed explanation can be found in Appendix A.7.2. From these DGMs $m$ synthetic data records are sampled that constitute the **default synthetic dataset** $S_1, ..., S_m$. We set $m = n$ to retain the dimensionality of the original data. Subsequently, each default synthetic dataset is debiased with respect to both estimands using the steps provided in Section 3, leading to a **debiased synthetic dataset**. Finally, two estimators are calculated in each of three datasets: the sample mean of *age* and the linear regression coefficient of *therapy* on *blood pressure* adjusted for

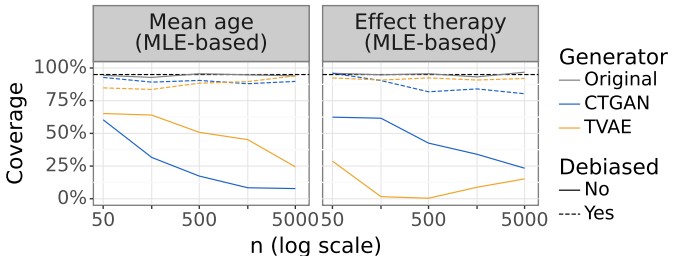

Figure 2: Empirical coverage of the 95% confidence interval for the population mean of *age* and the population effect of *therapy* on *blood pressure* adjusted for *stage*.

*stage*. We always report the maximum likelihood estimation (MLE)-based estimators, as used in traditional statistical analysis, of which the standard errors (SEs) are inflated with the correction factor $\sqrt{1 + m/n}$ to acknowledge the sampling variability of synthetic data. These estimators will deliver similar estimates as the EIC-based estimators since no data-adaptive estimation is used (see Section 3.3 and Appendix A.7.5).

## 4.2 Results

We now present the results of our simulation study. The DGMs were trained using the default hyperparameters as suggested by the package `Synthcity` (Qian et al., 2023). We also show results obtained for other hyperparameters (the default in the package `SDV` (Patki et al., 2016)) in Appendix A.7.4. In Section 4.2.1 we evaluate the empirical coverage of the 95% confidence interval (CI) for the population parameters. Our debiasing strategy should enhance the coverage, preferably to the nominal level, allowing for (more) honest inference, which is the main contribution of our strategy. Then, we analyse step by step the various components that may influence the coverage by investigating the bias and SE of the estimators in Section 4.2.2, and their convergence rates in Section 4.2.3. Additional results are presented in Appendix A.7.4, including the convergence rates of the nuisance parameters estimated by the DGMs. Our code is available on Github: `https://github.com/syndara-lab/debiased-generation`.

### 4.2.1 Coverage

By definition, 95% (*empirical*) of the 95% CIs (*nominal*) should cover the population parameter. Figure 2 depicts the empirical coverage of the 95% CIs obtained from both original and synthetic samples for the population mean of *age* and the population effect of *therapy* on *blood pressure* adjusted for *stage*. The results indicate that our debiasing strategy delivers empirical coverage levels for the population mean that approximate the nominal level for all sample sizes and DGMs considered. By contrast, the coverage based on the default synthetic datasets decreases with increasing $n$ due to slower shrinkage of the SE than the typical 1 over root-$n$ rate, as calculated in Section 4.2.3 and previously elaborated in Decruyenaere et al. (2024). For the population regression coefficient, debiasing delivers coverage at the nominal level for `TVAE` across all sample sizes, but not for `CTGAN`, although it clearly provides more honest inferences than based on the default synthetic datasets. The residual loss of coverage likely results from not using (efficient) sample splitting (see Appendix A.3).

### 4.2.2 Bias and Standard Error

Figures 3a and 3b depict the estimates and their SE, respectively, for the sample mean of *age* obtained in the default and debiased synthetic datasets. Figures A5 and A6 in the appendix show these for the linear regression coefficient of *therapy* on *blood pressure* adjusted for *stage*. In Figure 3a, each dot is an estimate per Monte Carlo run and the true population parameters are represented by the horizontal dashed line. This figure allows a qualitative assessment of two key properties of estimators: empirical bias (i.e., the average difference between the estimates and the population parameter, as represented by the solid line) and empirical SE (i.e., the standard deviation of the estimates, as indicated by the vertical spread of the estimates). Ideally, both converge to zero as the sample size grows larger. The convergence rate conveys the rate at which this happens. The funnel represents the default behaviour of an unbiased estimator based on original data of which the SE diminishes at a rate of 1 over root-$n$.

As shown in Figure 3a, the sample mean of *age* is unbiased in the default synthetic datasets, but exhibits a large empirical SE that shrinks slowly with sample size due to the data-adaptive nature of

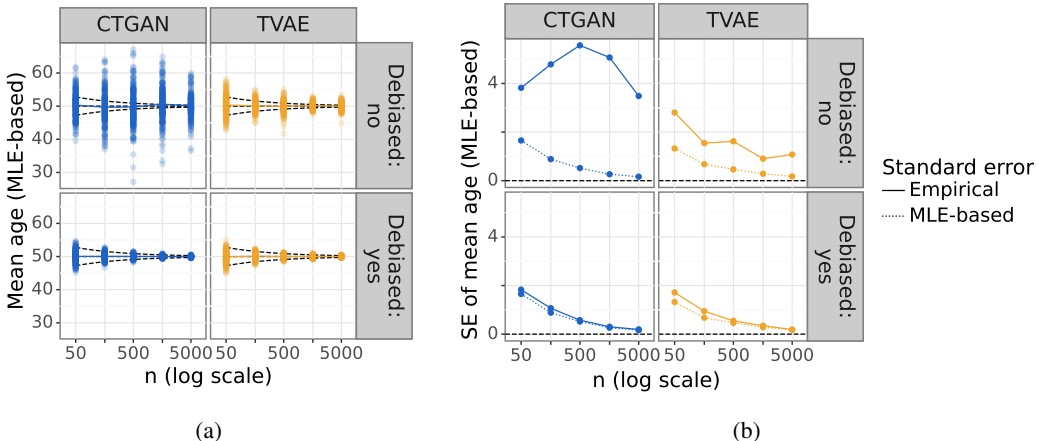

Figure 3: Each dot in Figure (a) is an estimate for the population mean of *age* per Monte Carlo run. The funnel indicates the behaviour of an unbiased and $\sqrt{n}$-consistent estimator based on observed data. Figure (b) depicts the empirical and average MLE-based SE for the sample mean of *age*.

DGMs. It is exactly this variability that is, on average, underestimated by the MLE-based SE in Figure 3b. Debiasing reduces the empirical variability and accelerates its shrinkage, such that the average MLE-based SE approximates the empirical SE. For the linear regression coefficient of *therapy* on *blood pressure* adjusted for *stage*, debiasing reduces finite-sample bias and also improves shrinkage of the empirical SE. Albeit less pronounced, the average MLE-based SE still underestimates the empirical SE with `CTGAN` despite debiasing (see Figures A5 and A6 in the appendix).

### 4.2.3 Convergence Rate of Standard Error

Assuming a power law $n^{-a}$ in convergence rate for the empirical SE, we estimate the exponent $a$ from five logarithmically spaced sample sizes $n$ between 50 and 5000, shown in Table 1 and Figure A7 in the appendix. Standard statistical analysis assumes that the bias converges faster than the SE with the latter diminishing at a rate of $1/\sqrt{n}$. However, the SEs produced by DGMs converge much slower (i.e., $a_{SE} < 0.5$), leading to a progressively increasing underestimation of the empirical SE by the MLE-based SE (which assumes $\sqrt{n}$-convergence) as the sample size grows larger. In turn, this results in too narrow CIs and poor coverage, as observed in Section 4.2.1. By contrast, our debiasing strategy renders estimators of which the SE converges at approximately root-$n$ rates (i.e., $a_{SE} = 0.5$), explaining the improvement in coverage of the CIs as a result of debiasing.

Table 1: Estimated exponent $a$ [95% CI] for the convergence rate $n^{-a}$ for empirical SE.

| Estimator | CTGAN | TVAE | CTGAN | TVAE |
|---|---|---|---|---|
| | **Default synthetic datasets** | | **Debiased synthetic datasets** | |
| Mean age | 0.01 [-0.16; 0.18] | 0.21 [0.04; 0.39] | 0.50 [0.46; 0.54] | 0.47 [0.44; 0.50] |
| Effect therapy | 0.31 [0.22; 0.40] | 0.09 [-0.13; 0.31] | 0.43 [0.31; 0.55] | 0.46 [0.38; 0.55] |

## 5 Case studies

To illustrate our findings and highlight their implications for the applied researcher, we conduct two case studies. First, Section 5.1 transfers the framework from our simulation study to the International Stroke Trial (IST) dataset (Sandercock et al., 2011). Second, Section 5.2 describes whether analysis results from the Adult Census Income dataset (Becker and Kohavi, 1996) are similar to those obtained from the real data, when the sample size $m$ of the generated synthetic data is very large. In both case studies, estimated SEs in the default synthetic and debiased synthetic datasets are corrected by multiplying with factor $\sqrt{1 + m/n}$ to acknowledge the sampling variability of synthetic data.

### 5.1 International Stroke Trial

We adapt the framework discussed in Section 4.1 to the IST dataset, one of the biggest randomised trials in acute stroke research (Sandercock et al., 2011). The dataset with 19285 complete cases now

constitutes our *population*. We mimic different hypothetical settings where an institution only has access to a limited *sample* of observations, with the sample size $n$ varying between 50 and 5000. In order to easily share the data with other researchers, the institution generates a synthetic dataset with sample size $m$, where $m = n$. We repeated this process 100 times per sample size $n$ to be able to calculate the empirical coverage levels. For illustration purposes, we focus on the effect of aspirin on the outcome at 6 months and report the proportion of deaths for the two treatment arms (aspirin and no aspirin), and its corresponding risk difference. For each value of $n$, two default synthetic datasets were generated using both CTGAN and TVAE. Next, for each of these, we first split the default synthetic dataset by treatment, debias the data with relation to the population mean within each treatment arm, and then combine them back into one debiased synthetic dataset. We noticed that using the same hyperparameters as in the simulation study resulted in biased estimates, as can be seen in Figure A12 in the appendix. For this reason, we highlight the results obtained by training with the default hyperparameters suggested by the package SDV (Patki et al., 2016) instead.

One of the original research questions in Sandercock et al. (2011) was whether or not there is a difference in risk of death between the treatment arms. Suppose a researcher can repeatedly collect information on 500 subjects and uses original, default synthetic and debiased synthetic data to make an inferential statement about this risk difference. Figure 4 depicts the confidence intervals for the first 15 repetitions, with the vertical dashed lines representing the true risk difference of $-0.009$ as calculated based on our population (the full dataset). Should the researcher use the default synthetic data, they would falsely conclude (in 7 out of these 15 repetitions) that the risk is significantly different from $-0.009$, while using the debiased synthetic dataset basically eliminates this high number of false-positives (with all these 15 intervals containing the population parameter), as is the case in the original data as well. More results can be found in Appendix A.8.1.

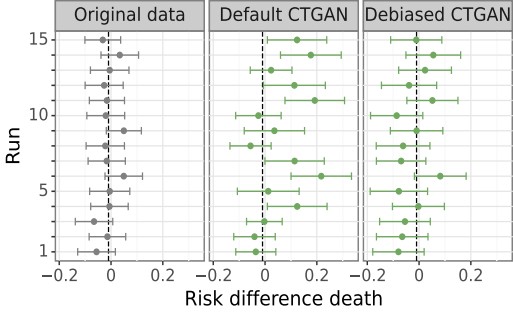

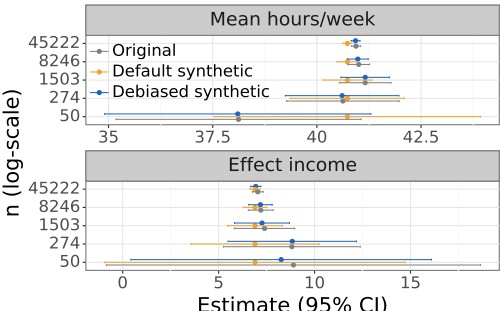

Figure 4: Empirical coverage of 95% CIs for the risk difference for *death* in each of the three datasets ($m = n = 500$) for the IST case study.

Figure 5: 95% CIs for the population mean and regression coefficient for $m = 10^6$ and different sample sizes $n$ for the Adult Income case study.

## 5.2 Adult Census Income Dataset

We also perform a case study on the Adult Census Income dataset, which comprises 45222 complete cases and 14 unique variables (Becker and Kohavi, 1996). We assume the researcher's interest lies in inferring the population mean of *hours* worked per week (estimated via the sample mean) and the average *sex*-adjusted difference in *age* between persons with an *income* of $> \$50$K a year vs. not (estimated via a linear regression model $age\,(Y) \sim income\,(A) + sex\,(X)$). Our goal in this case study is to confirm whether inferential results obtained using the debiased synthetic dataset are asymptotically equivalent (i.e. with $m >> n$ in our debiasing strategy) to those obtained using the original data. To test this across different sample sizes, the original data constitute five different samples of the Adult Census Income dataset with sizes $n$ varying log-uniformly between 50 and 45222. For each original dataset, a default synthetic dataset of size $m = 10^6$ was generated by TVAE. Subsequently, this dataset was debiased following the steps described in Section 3.1 (sample mean) and Section 3.2 (linear regression coefficient).

Figure 5 depicts the 95% CIs for both estimators, the five original sample sizes and the three versions of datasets. This indeed confirms that analysis on the debiased synthetic dataset leads to results of similar quality and CIs of similar length compared to the original dataset. By contrast, the analysis on the default synthetic dataset may yield results of inferior quality and even incorrect conclusions.

# 6 Discussion

In this paper, we propose a new debiasing strategy that targets synthetic data created by DGMs towards the downstream task of statistical inference from the resulting synthetic data. We establish our theory for two estimators by applying insights from debiased or targeted machine learning literature (van der Laan and Rose, 2011; Chernozhukov et al., 2018) to the current setting where machine learning is not necessarily used in the analysis, but rather in the generation of synthetic data. We obtain estimators that are less sensitive to the typical slow (statistical) convergence affecting DGMs and thereby improve the inferential utility of the synthetic data.

We illustrated the impact of our proposal through a simulation study on toy data and two case studies on real-world data. Our debiasing strategy results in root-$n$ consistent estimators based on the synthetic data and thereby better coverage of the confidence intervals, allowing for more honest inference. While coverage was clearly improved, it was not guaranteed to be at the nominal level. Indeed, it may remain anti-conservative for some estimators and DGMs, due to slow convergence inherent to these models and/or due to residual overfitting bias that could not be removed since sample splitting was not performed. Future work should focus on efficient sample splitting, where the resulting bias reduction outweighs the increase in finite-sample bias that arises from training on smaller sample sizes. Alternatively, findings from Ghalebikesabi et al. (2022) on importance weighting could be incorporated, with the weights being targeted to eliminate the impact of the data-adaptive estimation of the weights. This may potentially relax the fast baseline convergence assumption, and enable the same debiased synthetic data to be used for multiple downstream analyses.

A key advantage of our debiasing strategy is that it may deliver synthetic data created by DGMs of which the analysis is equivalent to the original data analysis, provided that the synthetic sample size is chosen to be much larger than the original sample size. Although the risk of disclosure may increase with the size of non-DP synthetic data (Reiter and Drechsler, 2010), this trade-off is beyond the scope of our paper. More interestingly, in the case of DP synthetic data, our debiasing strategy may exploit their post-processing immunity that allows for data transformations without compromising privacy guarantees (Dwork and Roth, 2014). However, our strategy needs to be extended to incorporate the DP-constraints when studying the difference between $\theta(\widetilde{P}_m)$ and $\theta(P)$ in DP synthetic data.

Limitations of our proposal include the low-dimensional setting of our simulation and case studies, for which DGMs might be less well suited. The positive results found for two widely used estimators in this simple setting highlight the utility of a debiasing approach and are encouraging in terms of future larger-scale applications. However, before addressing these, it is important to first understand low-dimensional settings, where valid inference is already challenging to attain. While our debiasing strategy boils downs to a post-processing step, one could argue that the lack of change to the DGM's training strategy itself is actually a strength, since it renders our strategy generator-agnostic.

Another limitation concerns the fact that our debiasing strategy for a regression coefficient requires sampling of synthetic data conditional on a covariate, which is not available in all DGMs. However, this issue is partially mitigated in the case of conditioning on categorical variables, since one can always generate a synthetic dataset unconditionally and then only select samples that fit the condition – though this approach also has its limits, especially when conditioning on multiple covariates at once. Zhou et al. (2023) propose a deep generative approach to sample from a conditional distribution, even when working with high-dimensional data. Future work could explore this strategy.

Finally, our proposal requires that the person generating synthetic data is aware of the analyses that will be run on those data, and has access to the corresponding EICs needed for debiasing (which in particular rules out the possibility for debiasing w.r.t. non-pathwise differentiable parameters, such as conditional means or predictions). For each parameter of interest, the data generated by the DGM will then need to be debiased (simultaneously) w.r.t. that parameter's EIC, which is left for future research. In the case of original data, several debiased estimation strategies that do not require exact knowledge about the EIC already exist. These methods include a) approximating the EIC through finite-differencing (Carone et al., 2019; Jordan et al., 2022) or stochastic approximations via Monte Carlo (Agrawal et al., 2024), and b) automatically estimating the EIC from the data through auto-DML (Chernozhukov et al., 2022). Alternatively, kernel debiased plug-in estimation methods (Cho et al., 2023) enable simultaneous debiasing of all pathwise differentiable target parameters that meet certain regularity conditions, without requiring any knowledge about the EIC. Integrating their insights could further strengthen the foundations of our current work on the interplay between synthetic data, deep generative modelling, and debiased machine learning.

## Acknowledgments and Disclosure of Funding

Paloma Rabaey's research is funded by the Research Foundation Flanders (FWO-Vlaanderen) with grant number 1170124N. This research also received funding from the Flemish government under the "Onderzoeksprogramma Artificiële Intelligentie (AI) Vlaanderen" programme.

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

# A Appendix

## A.1 Elaboration estimators for synthetic data

**Population mean.** The population mean of the observed data $o$ is defined as

$$\theta(P) = \int o \frac{dP(o)}{do} do = \int o dP(o),$$

which we will denote short as $\int OdP$ and its efficient influence curve (EIC) is $\phi(O, P) = O - \theta(P)$. Choosing $\widetilde{P}_m = \widetilde{\mathbb{P}}_m$, we will then study the large sample behaviour of the synthetic data estimator:

$$\theta(\widetilde{P}_m) = \int s d\widetilde{\mathbb{P}}_m(s) = \frac{1}{m} \sum_{i=1}^{m} S_i.$$

**Linear regression.** Suppose we are interested in a specific coefficient $\theta \equiv \theta(P)$ of some exposure $A$ in the linear model

$$E_P(Y|A, X) = \theta A + \omega(X),$$

where $X$ is a possibly high-dimensional covariate and $\omega(.)$ is an unknown function. Our proposal allows $\omega(X)$ to correspond to a standard linear model, but is less restrictive. The parameter $\theta$ in this linear model can be nonparametrically defined as

$$\theta(P) = \frac{E_P\left[\{A - E_P(A|X)\}\{Y - E_P(Y|X)\}\right]}{E_P\left[\{A - E_P(A|X)\}^2\right]}$$

in the sense that it reduces to $\theta$ when the above model holds. Its EIC is (Vansteelandt and Dukes, 2022)

$$\phi(O, P) = \frac{\{A - E_P(A|X)\}\left[Y - E_P(Y|X) - \theta(P)\{A - E_P(A|X)\}\right]}{E_P\left[\{A - E_P(A|X)\}^2\right]}$$

Denote the obtained synthetic data samples as $S = (\widetilde{Y}, \widetilde{A}, \widetilde{X})$. An estimator $\theta(\widetilde{P}_m)$ is then obtained by substituting in the above expression for $\theta(P)$, the first expectation in the numerator and denominator by a sample average, $E_P(A|X)$ and $E_P(Y|X)$ by data-adaptive predictions $E_{\widetilde{P}_m}(A|\widetilde{X}_i)$ and $E_{\widetilde{P}_m}(Y|\widetilde{X}_i)$ obtained based on the synthetic data. This delivers estimator

$$\theta(\widetilde{P}_m) = \frac{\frac{1}{m}\sum_{i=1}^{m}\left\{\widetilde{A}_i - E_{\widetilde{P}_m}(A|\widetilde{X}_i)\right\}\left\{\widetilde{Y}_i - E_{\widetilde{P}_m}(Y|\widetilde{X}_i)\right\}}{\frac{1}{m}\sum_{i=1}^{m}\left\{\widetilde{A}_i - E_{\widetilde{P}_m}(A|\widetilde{X}_i)\right\}^2}.$$

## A.2 Derivation of the impact of uncertainty affecting deep generative models

We are interested in knowing how much $\theta(\widetilde{P}_m)$ differs from $\theta(P)$. For this, we study the difference $\theta(\widetilde{P}_m) - \theta(P)$, for which we consider 2 von Mises expansions (i.e., functional Taylor expansions). Throughout the calculation below, we use that influence curves $\phi(O, P)$ have the property of being mean zero when averaging w.r.t. the distribution $P$. We moreover let $R(.)$ be a remainder term, which can generally be shown to be small, but must be studied on a case-by-case basis.

$$
\begin{aligned}
\theta(\widetilde{P}_m) - \theta(P) &= \theta(\widetilde{P}_m) - \theta(\widehat{P}_n) + \theta(\widehat{P}_n) - \theta(P) \\
&= \int \phi(S, \widetilde{P}_m) d(\widetilde{P}_m - \widehat{P}_n) + R(\widetilde{P}_m, \widehat{P}_n) \\
&\quad + \int \phi(O, \widehat{P}_n) d(\widehat{P}_n - P) + R(\widehat{P}_n, P) \\
&= - \int \phi(S, \widetilde{P}_m) d\widehat{P}_n + R(\widetilde{P}_m, \widehat{P}_n) - \int \phi(O, \widehat{P}_n) dP + R(\widehat{P}_n, P) \\
&= \int \phi(S, \widetilde{P}_m) d(\widetilde{\mathbb{P}}_m - \widehat{P}_n) - \int \phi(S, \widetilde{P}_m) d\widetilde{\mathbb{P}}_m + R(\widetilde{P}_m, \widehat{P}_n) \\
&\quad + \int \phi(O, \widehat{P}_n) d(\mathbb{P}_n - P) - \int \phi(O, \widehat{P}_n) d\mathbb{P}_n + R(\widehat{P}_n, P) \\
&= \int \phi(S, \widehat{P}_n) d(\widetilde{\mathbb{P}}_m - \widehat{P}_n) - \int \phi(S, \widetilde{P}_m) d\widetilde{\mathbb{P}}_m + R(\widetilde{P}_m, \widehat{P}_n) \\
&\quad + \int \left\{ \phi(S, \widetilde{P}_m) - \phi(S, \widehat{P}_n) \right\} d(\widetilde{\mathbb{P}}_m - \widehat{P}_n) \\
&\quad + \int \phi(O, P) d(\mathbb{P}_n - P) - \int \phi(O, \widehat{P}_n) d\mathbb{P}_n + R(\widehat{P}_n, P) \\
&\quad + \int \left\{ \phi(O, \widehat{P}_n) - \phi(O, P) \right\} d(\mathbb{P}_n - P) \\
&= \frac{1}{m} \sum_{i=1}^{m} \phi(S_i, \widehat{P}_n) - \frac{1}{m} \sum_{i=1}^{m} \phi(S_i, \widetilde{P}_m) + R(\widetilde{P}_m, \widehat{P}_n) \\
&\quad + \int \left\{ \phi(S, \widetilde{P}_m) - \phi(S, \widehat{P}_n) \right\} d(\widetilde{\mathbb{P}}_m - \widehat{P}_n) \\
&\quad + \frac{1}{n} \sum_{i=1}^{n} \phi(O_i, P) - \frac{1}{n} \sum_{i=1}^{n} \phi(O_i, \widehat{P}_n) + R(\widehat{P}_n, P) \\
&\quad + \int \left\{ \phi(O, \widehat{P}_n) - \phi(O, P) \right\} d(\mathbb{P}_n - P).
\end{aligned}
$$

Here,

$$
\frac{1}{n} \sum_{i=1}^{n} \phi(O_i, P)
$$

equals 1 over root-$n$ times a term that is asymptotically normal (as $n$ goes to infinity) with mean zero and variance $E\left\{ \phi(O, P)^2 \right\}$. Further, conditional on $\widehat{P}_n$,

$$
\frac{1}{m} \sum_{i=1}^{m} \phi(S_i, \widehat{P}_n)
$$

equals 1 over root-$m$ times a term that converges to $E\left\{ \phi(S, \widehat{P}_n)^2 | \widehat{P}_n \right\}^{1/2}$ times a standard normal distribution (as $m$ goes to infinity). This follows from the mean zero property of influence curves and the fact that the synthetic data are drawn from the distribution $\widehat{P}_n$. This mean zero property is essential as it implies that the variation of $\widehat{P}_n$ across repeated (observed data) samples does not

induce excess variability. Next letting also the sample size $n$ go to infinity, we obtain convergence of $m^{-1/2} \sum_{i=1}^{m} \phi(S_i, \widehat{P}_n)$ to $E\left\{\phi(O, P)^2\right\} N(0, 1)$ under the assumption that $E\left\{\phi(S, \widehat{P}_n)^2 | \widehat{P}_n\right\}$ converges in probability to $E\left\{\phi(O, P)^2\right\}$. This is a weak assumption because

$$E\left\{\phi(S, \widehat{P}_n)^2 | \widehat{P}_n\right\} = \int \phi(o, \widehat{P}_n)^2 d\widehat{P}_n(o)$$

is generally smooth (continuous) in the distribution of the data (as in the considered two examples). Moreover, the flexibility offered by deep generative models (DGMs) makes it reasonable to assume that $\widehat{P}_n$ converges to $P$ (e.g., in $L_2(P)$); while this convergence may be slow, no requirements on the rate of convergence are needed for the above assumption to be satisfied.

### A.3 Note on sample splitting

In order to let

$$\int \left\{ \phi(O, \widehat{P}_n) - \phi(O, P) \right\} d(\mathbb{P}_n - P)$$

converge to zero, we will need the DGM to be trained on a different part of the data than the one on which the debiasing step will be applied. Such sample splitting is needed to prevent overfitting bias that may otherwise result from the highly data-adaptive nature of DGMs. In addition, we need $\widehat{P}_n$ (e.g., the DGM) to consistently estimate $P$ in the sense that the squared mean (under $P$) of $\phi(O, \widehat{P}_n) - \phi(O, P)$ (at fixed $\widehat{P}_n$) converges to zero in probability.

To prevent efficiency loss with sample splitting, one may consider the use of $k$-fold cross-fitting. For this, we randomly split the data in $k$ folds, each time train the DGM on $k - 1$ folds to obtain an estimator of the observed data distribution $\widehat{P}_{(k-1)n/k}$ and calculate the bias

$$-\frac{1}{n/k} \sum_i \phi(O_i, \widehat{P}_{(k-1)n/k}),$$

based on the $n/k$ data points $O_i$ in the remaining fold. The average of the $k$ obtained bias estimates can next be used for debiasing (see the next section for specific examples).

However, it remains to be seen from future work if the resulting bias reduction outweighs the increase in finite-sample bias that may result from training the DGM on smaller sample sizes. Preliminary simulations with a simple implementation suggested that this was not the case, which is why our results in the main text are reported without the use of sample splitting. Furthermore, the remainder of the theory discards this nuance for now as well.

## A.4 Graphical summary

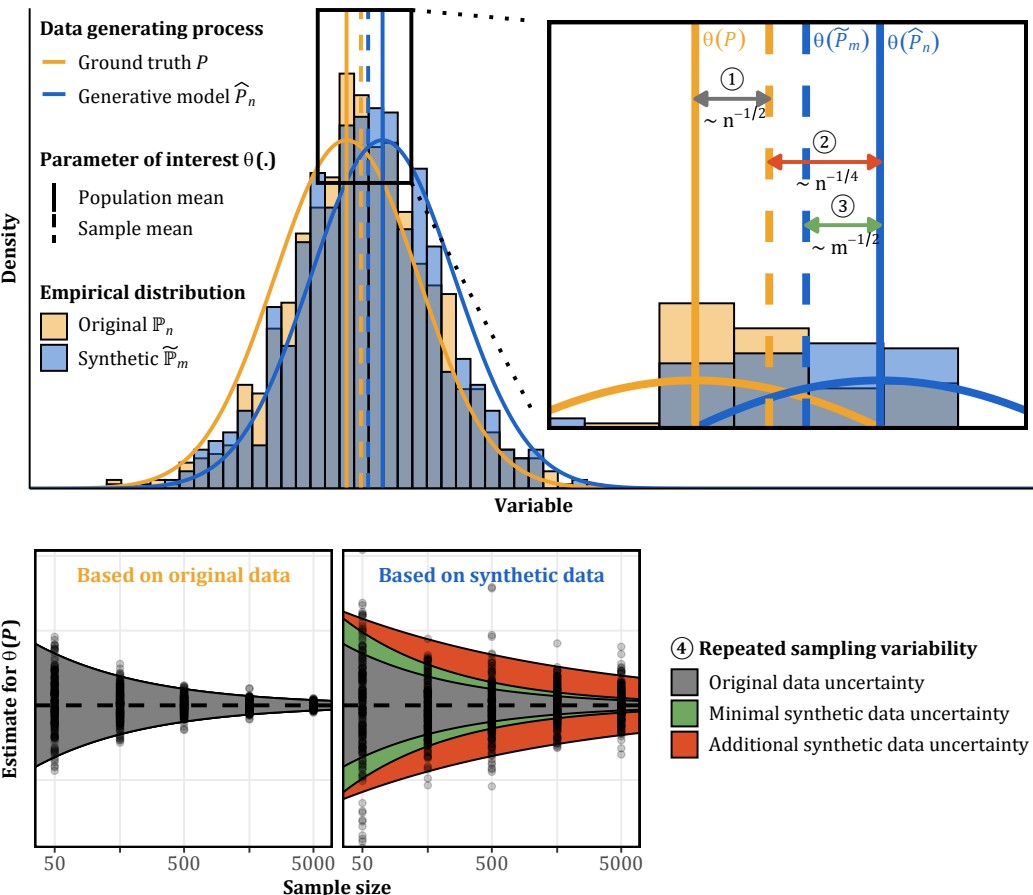

Figure A1: Suppose that interest lies in inferring the mean $\theta(.)$ (orange solid vertical line) of the ground truth distribution $P$ (orange solid curve) from synthetic data. **(1)** First, a random sample of original data of size $n$ with empirical distribution $\mathbb{P}_n$ (orange histogram) is collected from this ground truth. Theory on asymptotic linearity prescribes that the sample mean (orange dashed vertical line) will deviate from the population mean by order of 1 over root-$n$ (grey arrow). **(2)** Subsequently, a deep generative model is trained on these original data, yielding an estimated distribution $\widehat{P}_n$ (blue solid curve). Its mean (blue solid vertical line) may in turn differ from the original sample mean by an order larger than 1 over root-$n$ (red arrow), which is referred to as regularisation bias in Decruyenaere et al. (2024). **(3)** Finally, synthetic data of size $m$ (here $m = n$) are sampled from the estimated distribution $\widehat{P}_n$, forming the empirical distribution $\widetilde{\mathbb{P}}_m = \widetilde{P}_m$ (blue histogram with sample mean indicated by the blue dashed vertical line). The mean of both distributions $\widehat{P}_n$ and $\widetilde{P}_m$ will again differ by order of 1 over root-$m$ (green arrow). Ideally, the data analyst, who uses the synthetic data to estimate $\theta(P)$ by $\theta(\widetilde{P}_m)$, needs to take into account these three sources of random variability. **(4)** The large sample behaviour of the synthetic data estimator $\theta(\widetilde{P}_m)$ is depicted by repeating the above procedure multiple times across increasing sample sizes of $n = m$ and storing each estimate of the synthetic sample mean. Although the estimator remains unbiased for $\theta(P)$ (dashed line), its empirical standard error becomes larger than in the original data due to the additional sources of variability. However, the correction factor $\sqrt{1 + m/n}$ to the model-based standard error previously proposed by Raab et al. (2016) only captures the original data sampling variability (grey funnel) and a lower bound of the synthetic data sampling variability (green funnel), while the uncertainty associated with the regularisation bias (red funnel) remains unaccounted for. Since it cannot readily be expressed analytically, our debiasing strategy will eliminate the latter by shifting the mean of the distribution $\theta(\widehat{P}_n)$ estimated by the generative model towards the original sample mean (thereby removing the red arrow and funnel). Additionally, choosing synthetic sample sizes of $m \to \infty$ will shrink the synthetic data sampling variability (ultimately removing the green arrow and funnel), such that the synthetic data estimator exhibits similar large sample behaviour as in original data.

## A.5 Elaboration on debiased strategy for linear regression coefficient

For the linear regression coefficient case, this section shows how the samples generated by a given DGM need to adapted to eliminate the bias $b$, given by

$$b = \frac{\sum_{i=1}^{n} \left\{ A_i - E_{\widehat{P}_n}(A|X_i) \right\} \left[ Y_i - E_{\widehat{P}_n}(Y|X_i) - \theta(\widehat{P}_n) \left\{ A_i - E_{\widehat{P}_n}(A|X_i) \right\} \right]}{\sum_{i=1}^{n} \left\{ A_i - E_{\widehat{P}_n}(A|X_i) \right\}^2}$$

$$= \frac{\sum_{i=1}^{n} \left\{ A_i - E_{\widehat{P}_n}(A|X_i) \right\} \left\{ Y_i - E_{\widehat{P}_n}(Y|X_i) \right\}}{\sum_{i=1}^{n} \left\{ A_i - E_{\widehat{P}_n}(A|X_i) \right\}^2} - \theta(\widehat{P}_n)$$

To compute this bias, we must generate, for each observed value $X_i$, a very large sample of measurements of $A$ and $Y$ with the given level $X_i$, based on the DGM. Then $E_{\widehat{P}_n}(Y|X_i)$ can be calculated as the sample average of those values for $Y$, and likewise $E_{\widehat{P}_n}(A|X_i)$ can be calculated as the sample average of those values for $A$. Further based on a large sample generated based on the DGM, we calculate $\theta(\widehat{P}_n)$ as the sample average of $\left\{ A - E_{\widehat{P}_n}(A|X) \right\} \left\{ Y - E_{\widehat{P}_n}(Y|X) \right\}$ divided by the sample average of $\left\{ A - E_{\widehat{P}_n}(A|X) \right\}^2$.

Debiasing of the DGM can now be done by adding $b\{\widetilde{A}_i - E_{\widehat{P}_n}(A|\widetilde{X}_i)\}$ to the synthetic outcome observations generated by the DGM. This change does not affect the predictions $E_{\widehat{P}_n}(Y|X_i)$ from the DGM (because $\widetilde{A}_i - E_{\widehat{P}_n}(A|X_i)$ averages to zero for each choice of $X_i$). Further, with this change, $\theta(\widehat{P}_n)$ also increases with $b$ units as follows:

$$\frac{\sum_{i=1}^{N} \left\{ \widetilde{A}_i - E_{\widehat{P}_n}(A|\widetilde{X}_i) \right\} \left\{ \widetilde{Y}_i + b \left\{ \widetilde{A}_i - E_{\widehat{P}_n}(A|\widetilde{X}_i) \right\} - E_{\widehat{P}_n}(Y|\widetilde{X}_i) \right\}}{\sum_{i=1}^{N} \left\{ \widetilde{A}_i - E_{\widehat{P}_n}(A|\widetilde{X}_i) \right\}^2},$$

where the sum runs over a large sample of synthetic observations. With this change in $\theta(\widehat{P}_n)$, the previously calculated bias becomes

$$\frac{\sum_{i=1}^{n} \left\{ A_i - E_{\widehat{P}_n}(A|X_i) \right\} \left\{ Y_i - E_{\widehat{P}_n}(Y|X_i) \right\}}{\sum_{i=1}^{n} \left\{ A_i - E_{\widehat{P}_n}(A|X_i) \right\}^2} - \left\{ \theta(\widehat{P}_n) + b \right\}$$

which is zero, and as such the debiasing with respect to term (6) is complete. Based on a new sample of synthetic observations (independent of those generated to calculate the bias), the synthetic data estimator is

$$\theta(\widetilde{P}_m) = \frac{\frac{1}{m}\sum_{i=1}^{m} \left\{ \widetilde{A}_i - E_{\widetilde{P}_m}(A|\widetilde{X}_i) \right\} \left[ \widetilde{Y}_i + b \left\{ \widetilde{A}_i - E_{\widetilde{P}_m}(A|\widetilde{X}_i) \right\} - E_{\widetilde{P}_m}(Y|\widetilde{X}_i) \right]}{\frac{1}{m}\sum_{i=1}^{m} \left\{ \widetilde{A}_i - E_{\widetilde{P}_m}(A|\widetilde{X}_i) \right\}^2}$$

$$= \frac{\frac{1}{m}\sum_{i=1}^{m} \left\{ \widetilde{A}_i - E_{\widetilde{P}_m}(A|\widetilde{X}_i) \right\} \left\{ \widetilde{Y}_i - E_{\widetilde{P}_m}(Y|\widetilde{X}_i) \right\}}{\frac{1}{m}\sum_{i=1}^{m} \left\{ \widetilde{A}_i - E_{\widetilde{P}_m}(A|\widetilde{X}_i) \right\}^2} + b.$$

This ensures that bias term (5) equals zero. Please note that this estimator coincides with the standard linear regression estimator on the debiased synthetic data when least squares predictions for $E_{\widetilde{P}_m}(A|\widetilde{X}_i)$ and $E_{\widetilde{P}_m}(Y|\widetilde{X}_i)$ are used.

## A.6 Derivation of variance of debiased estimator $\theta(\widetilde{P}_m)$

With the suggested debiasing, we thus expect that

$$\theta(\widetilde{P}_m) - \theta(P) \quad = \quad \frac{1}{m} \sum_{i=1}^{m} \phi(S_i, \widehat{P}_n) + \frac{1}{n} \sum_{i=1}^{n} \phi(O_i, P) + o_p(n^{-1/2}) + o_p(m^{-1/2}).$$

Since the synthetic data are independently drawn from the observed data, any covariance between the 2 leading terms must originate from the fact that $\widehat{P}_n$ depends on the observed data. This covariance equals zero since

$$
\begin{aligned}
E\left\{\phi(S_i, \widehat{P}_n)\phi(O_j, P)\right\} &= E\left[E\left\{\phi(S_i, \widehat{P}_n)|O_1, ..., O_n\right\}\phi(O_j, P)\right] \\
&= E\left[E\left\{\phi(S_i, \widehat{P}_n)|\widehat{P}_n, O_1, ..., O_n\right\}\phi(O_j, P)\right] \\
&= E\left[E\left\{\phi(S_i, \widehat{P}_n)|\widehat{P}_n\right\}\phi(O_j, P)\right] = 0
\end{aligned}
$$

where in the second equality we use that $\widehat{P}_n$ is determined by $O_1, ..., O_n$, in the third equality we use that $S_i$ only depends on $O_1, ..., O_n$ via $\widehat{P}_n$, and in the final equality we use that $\phi(S_i, \widehat{P}_n)$ has mean zero when the synthetic data are sampled from $\widehat{P}_n$. This renders these terms asymptotically independent and their sum, hence, asymptotically normal. Moreover, since the variance of $\phi(S_i, \widehat{P}_n)$ converges to $E\left\{\phi(O, P)^2\right\}$ (see the previous section; Appendix A.2), the variance of $\theta(\widetilde{P}_m)$ may thus be approximated by

$$\left(\frac{1}{m} + \frac{1}{n}\right) E\left\{\phi(O, P)^2\right\},$$

thereby generalising results known for parametric synthetic data generators (Raab et al., 2016; Decruyenaere et al., 2024).

### A.7 Simulation study

#### A.7.1 Data generating process

---

**Algorithm 1:** Data generating process for hypothetical disease.

---

**input** : Requested number of data records $n$.
**output** : Dataframe $D$ with $n$ records, each made up of 4 attributes: $age, stage, therapy, bp$.

$D \leftarrow Empty\ dataframe$
**for** $i \leftarrow 1$ **to** $n$ **do**

    $age \leftarrow Normal(mean = 50, std = 10)$

    $\nu_{age} \leftarrow 0.05$
    $\nu_I, \nu_{II}, \nu_{III} \leftarrow 2, 3, 4$
    $cp_I, cp_{II}, cp_{III} \leftarrow$
      $Sigmoid(\nu_I - \nu_{age} \times age), Sigmoid(\nu_{II} - \nu_{age} \times age), Sigmoid(\nu_{III} - \nu_{age} \times age)$
    $stage \leftarrow Categorical(cat = [I, II, III, IV], probs = [cp_I, cp_{II} - cp_I, cp_{III} - cp_{II}, 1 - cp_{III}])$

    $therapy \leftarrow Categorical(cat = [False, True], p = [0.5, 0.5])$

    $\beta_{therapy} \leftarrow -20$
    $\beta_I, \beta_{II}, \beta_{III}, \beta_{IV} \leftarrow 0, 10, 20, 30$
    $\mu_{bp} \leftarrow 120 + \beta_{stage} + \beta_{therapy} \times therapy$
    $bp \leftarrow Normal(mean = \mu_{bp}, std = 10)$

    $D_i \leftarrow \{age, stage, therapy, bp\}$
**end**

---

Inspired by an applied medical setting, we create a hypothetical disease, defined by a low-dimensional tabular data generation process. The dependency structure depicted by the directed acyclic graph (DAG) in Figure 1 in the main text displays the presence of four variables, each of them chosen to obtain a mix of data types. In our hypothetical disease, it is assumed that a patient is observed at a given point in time. At this time, patient data about *age*, atherosclerosis *stage*, and the random assignment of *therapy* is gathered. The continuous outcome variable *blood pressure* is evaluated at a later time point, making this design a simplification, since we do not consider the data as longitudinal.

The exact routine to reconstruct this data generating process is presented in the pseudo-code in Algorithm 1. *Age* (continuous) follows a normal distribution with mean 50 and standard deviation 10. Atherosclerosis *stage* (ordinal) was generated according to a proportional odds cumulative logit model where an increase in *age* causes an increase in the odds of having a *stage* higher than a given stage $k$ ($\nu_{age} = -0.05$ and intercepts $\nu_{stage} = \{2, 3, 4\}$ for stage I-III). *Therapy* (binary) is considered to be 1:1 randomly assigned and is therefore sampled from a Bernouilli distribution with a probability of $0.50$. The last variable, *blood pressure* (continuous), is sampled from a normal distribution with standard deviation 10 and where the baseline average of 120 increases with higher atherosclerosis *stage* ($\beta_{stage} = \{0, 10, 20, 30\}$ for stage I-IV, respectively) and absence of *therapy* ($\beta_{therapy} = -20$).

#### A.7.2 Deep Generative Models

We elaborate on the DGMs used to create synthetic data in our simulation study and cases studies. All experiments were run on our institutional high performance computing cluster using a single GPU (NVIDIA Ampere A100; 80GB GPU memory) and single CPU (AMD EPYC 7413), taking less than 24 hours to complete (simulation study: less than 15 minutes per individual run across 5 sample sizes; International Stroke Trial case study: less than 75 minutes per individual run across 5 sample sizes; Adult Census Income Dataset case study: less than 4 hours). We focus on two commonly used DGMs, namely Generative Adversarial Networks (GANs) (Goodfellow et al., 2014) and Variational Autoencoders (VAEs) (Kingma and Welling, 2013).

A GAN consists of two competing neural networks, a generator and discriminator, and aims to achieve an equilibrium between both (Hernandez et al., 2022). This translates to a mini-max game, since the generator aims to minimise the difference between the real and generated data, while the discriminator aims to maximise the possibility to distinguish the real and generated data (Goodfellow

et al., 2014). We use the `CTGAN` implementation that was designed specifically for tabular data, proposed by Xu et al. (2019).

A VAE is a deep latent variable model, consisting of an encoder and a decoder (Kingma and Welling, 2013). The encoder models the approximate posterior distribution of the latent variables given an input instance, whereby typically a standard normal prior is assumed for the latent variables. The decoder allows reconstructing an input instance, based on a sample from the predicted latent space distribution. Encoder and decoder can be jointly trained by maximising the evidence lower bound (ELBO), i.e. the marginal likelihood of the training instances. Maximising the ELBO corresponds to minimising the Kullback-Leibler (KL) divergence between the predicted latent variable distribution for a given input instance and the standard normal priors, and minimising the reconstruction error of the input instance at the decoder output. Once again, we use the tabular implementation of a VAE (`TVAE`) proposed by Xu et al. (2019).

The results presented in the rest of the Appendix are obtained by training both types of DGM with default hyperparameters as suggested by the packages `Synthcity` and `SDV` (for both `CTGAN` and `TVAE`). A comparison of the default hyperparameters in both packages is provided in Tables A1 and A2. Note that both packages implement the `CTGAN` and `TVAE` modules as originally proposed by Xu et al. (2019), where a cluster-based normaliser is used to preprocess numerical features.

Table A1: Comparison of `CTGAN` default hyperparameters between `SDV` and `Synthcity`.

| Hyperparameter | SDV | Synthcity |
|---|---|---|
| generator_dim | (256, 256) | (500, 500) |
| discriminator_dim | (256, 256) | (500, 500) |
| generator_dropout | 0.0 | 0.1 |
| discriminator_dropout | 0.5 | 0.1 |
| generator_lr | $2 \times 10^{-4}$ | $1 \times 10^{-3}$ |
| generator_decay | $1 \times 10^{-6}$ | $1 \times 10^{-3}$ |
| discriminator_lr | $2 \times 10^{-4}$ | $1 \times 10^{-3}$ |
| discriminator_decay | $1 \times 10^{-6}$ | $1 \times 10^{-3}$ |
| batch_size | 500 | 200 |
| discriminator_steps | 1 | 1 |
| epochs | 300 | 2000 |

Table A2: Comparison of `TVAE` default hyperparameters between `SDV` and `Synthcity`.

| Hyperparameter | SDV | Synthcity |
|---|---|---|
| embedding_dim | 128 | 500 |
| encoder_dim | (128, 128) | (500, 500, 500) |
| decoder_dim | (128, 128) | (500, 500, 500) |
| encoder_dropout | 0.0 | 0.1 |
| decoder_dropout | 0.0 | 0.1 |
| lr | $1 \times 10^{-3}$ | $1 \times 10^{-3}$ |
| decay | $1 \times 10^{-5}$ | $1 \times 10^{-5}$ |
| loss_factor | 2 | 1 |
| batch_size | 500 | 200 |
| epochs | 300 | 1000 |

### A.7.3 Quality of synthetic data

We performed some additional analyses to assess the quality of the synthetic data obtained in our simulation study.

**Average IKLD.** The inverse of the KL divergence (IKLD) between original and synthetic data, averaged over 250 Monte Carlo runs and standardised between 0 and 1, is presented in Table A3, where the debiased synthetic datasets have slightly higher IKLD than their default versions for all DGMs considered.

Table A3: The IKLD between original and synthetic data, averaged over 250 Monte Carlo runs and standardised between 0 and 1. Higher values indicate similar datasets in terms of underlying distribution.

| Generator | Debiased | $n = 50$ | $n = 160$ | $n = 500$ | $n = 1600$ | $n = 5000$ |
|---|---|---|---|---|---|---|
| **CTGAN (SDV)** | **no** | 0.849 | 0.924 | 0.939 | 0.963 | 0.980 |
| **CTGAN (SDV)** | **yes** | 0.860 | 0.935 | 0.957 | 0.966 | 0.981 |
| **CTGAN (Synthcity)** | **no** | 0.838 | 0.894 | 0.867 | 0.890 | 0.929 |
| **CTGAN (Synthcity)** | **yes** | 0.849 | 0.906 | 0.885 | 0.897 | 0.933 |
| **TVAE (SDV)** | **no** | 0.704 | 0.819 | 0.908 | 0.985 | 0.980 |
| **TVAE (SDV)** | **yes** | 0.734 | 0.832 | 0.913 | 0.987 | 0.983 |
| **TVAE (Synthcity)** | **no** | 0.791 | 0.830 | 0.887 | 0.929 | 0.977 |
| **TVAE (SDV)** | **yes** | 0.813 | 0.860 | 0.913 | 0.943 | 0.980 |

**Failed generators.** `CTGAN (Synthcity)` could not be trained in 14 runs (runs $38, 69, 102, 225$ for $n = 500$; runs $19, 23, 98, 107, 117, 128, 129, 140, 148, 169$ for $n = 5000$) due to an internal error in the package. As such, it was not possible to generate default and debiased synthetic data with `CTGAN (Synthcity)` in these runs. This comprises $1.12\%$ $(14/1250)$ of all `CTGAN (Synthcity)` trained and $0.28\%$ $(14/5000)$ of all generators trained. The other generative models did not produce errors during training, so that default synthetic data could be generated in every run.

**Failed debiasing.** `TVAE (SDV)` could not be debiased in $82$ runs (runs $1, 3, 4, 5, 6, 8,$ $15, 23, 24, 28, 30, 42, 46, 50, 51, 58, 64, 72, 78, 92, 96, 104, 106, 111, 112, 117, 118, 119, 122, 126,$ $127, 130, 133, 135, 140, 143, 146, 149, 154, 157, 159, 160, 166, 167, 168, 170, 178, 180, 187, 191,$ $193, 196, 197, 200, 205, 210, 214, 216, 218, 220, 225, 230, 231, 235, 236, 245, 247, 248$ for $n = 50$; runs $13, 22, 97, 117, 139, 145, 146, 148, 161, 186, 193, 197, 235, 246$ for $n = 160$) due to sparse data, especially for small sample sizes. In particular, some $X_i$ in the original dataset may not be present in the default synthetic dataset generated by `TVAE (SDV)`, leading to an error when estimating the nuisance parameters needed for the debiasing step. This comprises $6.56\%$ $(82/1250)$ of all `TVAE (SDV)` trained and $1.64\%$ $(82/5000)$ of all generators trained. The other generative models did not produce errors during debiasing, so that debiased synthetic data could be generated in every run.

**Exact memorisation.** A sanity check was conducted to ensure that no records of the original data were memorised by the generative model. `CTGAN (Synthcity)` made the following number of exact copies of the original data in the synthetic data: one record $(2.00\%)$ for $n = 50$ in three runs (runs $63, 158, 192$). The other generative models did not make exact copies.

**Non-estimable estimators.** Due to sparse data, especially for small sample sizes, the linear regression coefficient of *therapy* on *blood pressure* adjusted for *stage* could not be estimated in a small subset of the $11140$ (original and synthetic) datasets, producing extremely small $(< 1\mathrm{e}{-10})$ or large $(> 1\mathrm{e}2)$ standard errors. Overall, $0.09\%$ $(10/11140)$ regression coefficient estimates could not be obtained: in 4 runs for default synthetic dataset of size $m = 50$ generated by `CTGAN (SDV)` and in 1 run for default synthetic dataset of size $m = 160$ generated by `CTGAN (Synthcity)`, and for their corresponding debiased synthetic datasets. The sample mean of *age* was always estimable.

### A.7.4 Additional results

While the main text only reports results using the `Synthcity` default hyperparameters, we additionally wanted to report a wider variety of trained models, without manually tuning anything, which is why results for `SDV` are presented here as well.

**Coverage for all estimators and models.** The results shown in Figure A2 indicate that our debiasing strategy delivers empirical coverage levels for the population mean that approximate the nominal level for all sample sizes and DGMs considered. By contrast, the coverage in the default synthetic datasets decreases with increasing $n$ due to slower shrinkage of the standard error (SE) than the typical 1 over root-$n$ rate, as calculated in Section 4.2.3 and previously addressed in Decruyenaere

et al. (2024). While our debiasing strategy clearly improves the empirical coverage for the population regression coefficient, it does, however, not guarantee this to be at the nominal level for all sample sizes and DGMs considered. In particular, debiasing only seems to achieve coverage at the nominal level for `TVAE (Synthcity)` across all sample sizes and for `TVAE (SDV)` at sufficiently large sample sizes. Our approach still falls short for `CTGAN (Synthcity)` and `CTGAN (SDV)`, although providing more honest inference than the default synthetic datasets.

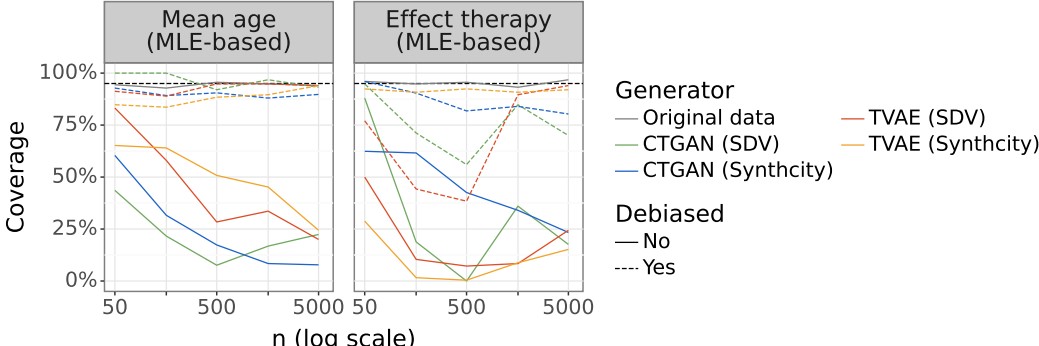

Figure A2: Empirical coverage of the 95% confidence interval for the MLE-based estimators.

**Bias and SE for mean *age*.** As shown in Figure A3, the sample mean of *age* is unbiased in the default synthetic datasets, but exhibits large variability that shrinks slowly with sample size due to the data-adaptive nature of DGMs. Debiasing reduces this variability and accelerates shrinkage. Figure A4 indicates that the empirical SE for the sample mean of *age* is indeed, on average, underestimated by the MLE-based SE in default synthetic datasets. After debiasing, the average MLE-based SE approximates the empirical SE, albeit with minor deviations at smaller sample sizes.

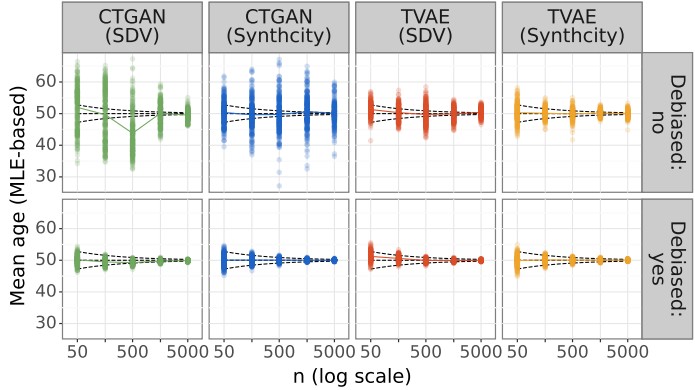

Figure A3: The horizontal dashed line represents the population mean of *age* and each dot is a MLE-based estimate per Monte Carlo run (250 dots in total per value of $n$). The dashed funnel indicates the behaviour of an unbiased and $\sqrt{n}$-consistent estimator based on observed data.

**Bias and SE for effect *therapy* on *blood pressure* adjusted for *stage*.** It can be seen from Figure A5 that the linear regression coefficient of *therapy* on *blood pressure* adjusted for *stage* in the default synthetic datasets has finite-sample bias towards the null that converges to zero as the sample size grows larger. Additionally, its variability seems to diminish slower than expected. Debiasing reduces the finite-sample bias and improves the shrinkage of the SE. However, Figure A6 shows that the average MLE-based SE in the debiased synthetic datasets still underestimates the empirical SE with `CTGAN (SDV)` and `CTGAN (Synthcity)` despite debiasing, albeit less pronounced.

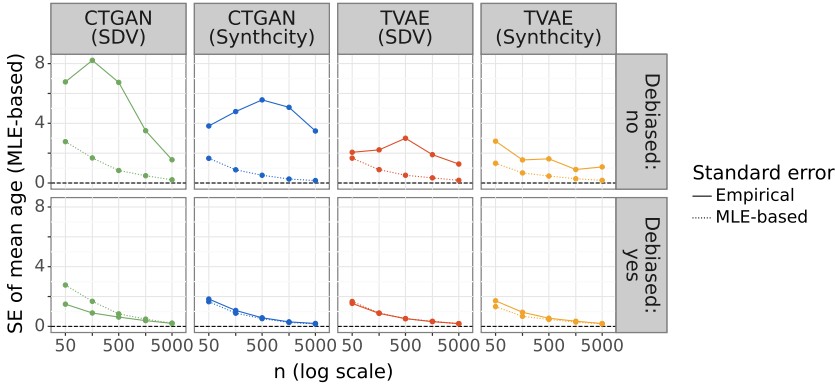

Figure A4: The empirical and MLE-based standard error for the sample mean of *age*.

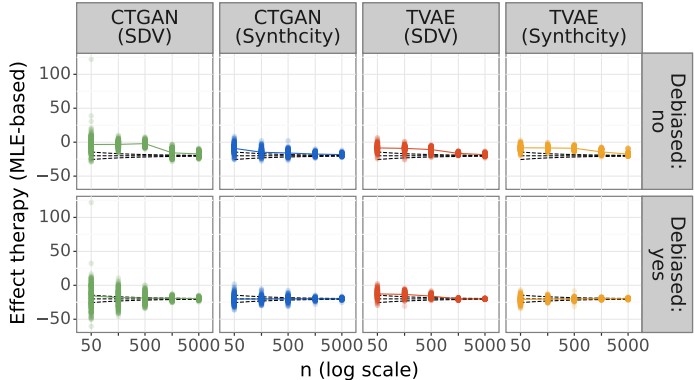

Figure A5: The horizontal dashed line represents the population effect of *therapy* on *blood pressure* adjusted for *stage* and each dot is a MLE-based estimate per Monte Carlo run (250 dots in total per value of $n$). The dashed funnel indicates the behaviour of an unbiased and $\sqrt{n}$-consistent estimator based on observed data.

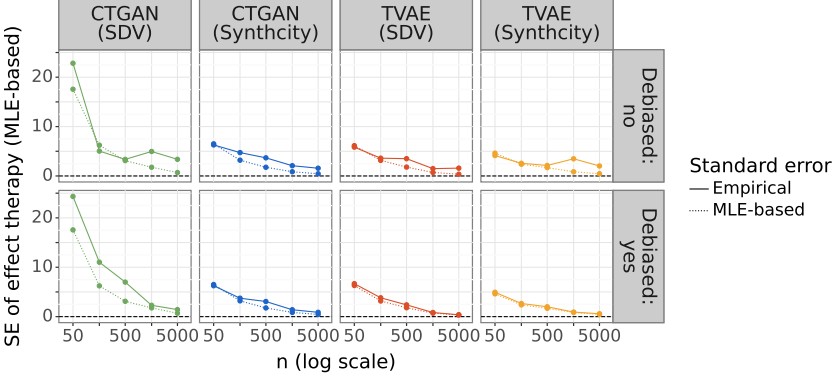

Figure A6: The empirical and MLE-based standard error for the (MLE-based) effect of *therapy* on *blood pressure* adjusted for *stage*.

**Convergence of SE for all estimators and models.** The convergence rate of the empirical SE for the MLE-based estimators is shown in Table A4 and represented by the slope in Figure A7. The dashed line indicates the behaviour of the SE of an unbiased and $\sqrt{n}$-consistent estimator based on observed data, whereas the dotted line indicates the assumed behaviour of the SE of the same estimator based on synthetic data, following the correction proposed by Raab et al. (2016). Lines for the empirical SE that are parallel to these lines indicate that the estimator converges at 1 over root-$n$ rate, whereas more horizontal or vertical lines imply slower or faster shrinkage, respectively. After debiasing, all lines are parallel for both estimators, suggesting $\sqrt{n}$-consistency. Note that the vertical offset of some lines, which represents the log asymptotic variance, is still too large for some DGMs despite debiasing. This is the case for the linear regression coefficient for *therapy* on *blood pressure* adjusted for *stage* obtained in the debiased synthetic dataset generated by `CTGAN (SDV)` and `CTGAN (SDV)`.

Table A4: Estimated exponent $a$ [95% CI] for the power law convergence rate $n^{-a}$ for empirical SE.

| Estimator | CTGAN (SDV) | CTGAN (Synthcity) | TVAE (SDV) | TVAE (Synthcity) |
|---|---|---|---|---|
| | **Default synthetic datasets** | | | |
| Mean age | 0.33 [0.04; 0.62] | 0.01 [-0.16; 0.18] | 0.10 [-0.13; 0.32] | 0.21 [0.04; 0.39] |
| Effect therapy | 0.23 [-0.09; 0.56] | 0.31 [0.22; 0.40] | 0.28 [0.10; 0.46] | 0.09 [-0.13; 0.31] |
| | **Debiased synthetic datasets** | | | |
| Mean age | 0.42 [0.36; 0.48] | 0.50 [0.46; 0.54] | 0.45 [0.44; 0.47] | 0.47 [0.44; 0.50] |
| Effect therapy | 0.58 [0.43; 0.73] | 0.43 [0.31; 0.55] | 0.61 [0.43; 0.79] | 0.46 [0.38; 0.55] |

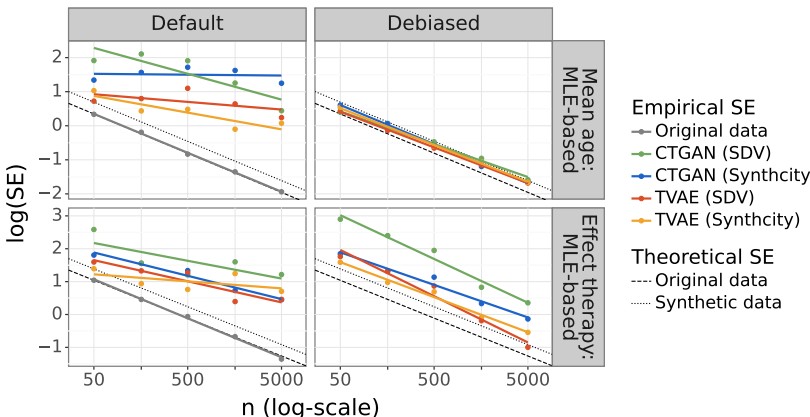

Figure A7: Convergence rate of the empirical standard error (SE) for the MLE-based estimators. If the SE is of the form $SE = cn^{-a}$, where $c$ is a constant, then $\log(SE) = \log(c) + (-a)\log(n)$. Therefore slope $a$ represents the convergence rate and the vertical offset $\log(c)$ indicates the log asymptotic variance. The dashed line indicates the behaviour of the SE of an unbiased and $\sqrt{n}$-consistent estimator based on observed data, whereas the dotted line indicates the assumed behaviour of the SE of the same estimator based on synthetic data, following the correction proposed by Raab et al. (2016).

**Nuisance parameters.** Our debiasing strategy assumes that the two remainder terms in the von Mises expansion presented in Section 3 are $o_p(m^{-1/2})$ and $o_p(n^{-1/2})$. This requires the difference between $\widetilde{P}_m$ (i.e., estimate of $\widehat{P}_n$ based on the sampled synthetic dataset of $m$ records) and $\widehat{P}_n$ (i.e., estimate of $P$ by the DGM trained on $n$ original records) and the difference between $\widehat{P}_n$ and $P$ (i.e., the data generating process), to converge to zero in $L_2(\widehat{P}_n)$ and $L_2(P)$, respectively, at faster than $n$-to-the-quarter convergence rates. While this holds true for the convergence of $\widetilde{P}_m$ to $\widehat{P}_n$ (see Table A5), this seems not always strictly attained for the functionals of $\widehat{P}_n$ with respect to $P$ that appear in the efficient influence curves for the regression coefficient (see Table A6), in particular for `CTGAN`

(Synthcity) and, to a lesser extent, for `CTGAN` (`SDV`) and `TVAE` (`Synthcity`). Nevertheless, a root-$n$ consistent estimator was still obtained after debiasing these DGMs (see Table A4).

Table A5: Estimated exponent $a$ [95% CI] for the power law convergence rate $n^{-a}$ for the difference between functionals of $\widetilde{P}_m$ and $\widehat{P}_n$ in $L_2(\widehat{P}_n)$.

| Functional | CTGAN (SDV) | CTGAN (Synthcity) | TVAE (SDV) | TVAE (Synthcity) |
|---|---|---|---|---|
| $E[A\|X=I]$ | 0.47 [0.41; 0.52] | 0.50 [0.44; 0.56] | 0.45 [0.39; 0.50] | 0.54 [0.44; 0.63] |
| $E[A\|X=II]$ | 0.46 [0.40; 0.52] | 0.46 [0.37; 0.55] | 0.58 [0.39; 0.78] | 0.50 [0.45; 0.54] |
| $E[A\|X=III]$ | 0.46 [0.35; 0.57] | 0.49 [0.37; 0.61] | 0.49 [0.21; 0.77] | 0.55 [0.52; 0.58] |
| $E[A\|X=IV]$ | 0.43 [0.37; 0.50] | 0.48 [0.42; 0.54] | 0.42 [0.32; 0.53] | 0.49 [0.44; 0.55] |
| $E[Y\|X=I]$ | 0.50 [0.34; 0.66] | 0.53 [0.48; 0.58] | 0.43 [0.35; 0.50] | 0.47 [0.36; 0.57] |
| $E[Y\|X=II]$ | 0.50 [0.35; 0.64] | 0.48 [0.36; 0.60] | 0.60 [0.30; 0.89] | 0.44 [0.40; 0.48] |
| $E[Y\|X=III]$ | 0.52 [0.41; 0.63] | 0.51 [0.35; 0.67] | 0.59 [0.24; 0.95] | 0.50 [0.45; 0.56] |
| $E[Y\|X=IV]$ | 0.51 [0.42; 0.61] | 0.50 [0.44; 0.55] | 0.56 [0.49; 0.63] | 0.42 [0.35; 0.50] |

Table A6: Estimated exponent $a$ [95% CI] for the power law convergence rate $n^{-a}$ for the difference between functionals of $\widehat{P}_n$ and $P$ in $L_2(P)$.

| Functional | CTGAN (SDV) | CTGAN (Synthcity) | TVAE (SDV) | TVAE (Synthcity) |
|---|---|---|---|---|
| $E[A\|X=I]$ | 0.24 [0.12; 0.36] | 0.16 [0.02; 0.30] | 0.46 [0.08; 0.84] | 0.30 [0.25; 0.35] |
| $E[A\|X=II]$ | 0.19 [-0.01; 0.38] | 0.22 [0.08; 0.36] | 0.64 [0.25; 1.04] | 0.32 [0.24; 0.40] |
| $E[A\|X=III]$ | 0.18 [-0.03; 0.40] | 0.22 [0.08; 0.36] | 0.51 [0.09; 0.94] | 0.27 [0.15; 0.39] |
| $E[A\|X=IV]$ | 0.19 [0.01; 0.38] | 0.23 [0.08; 0.38] | 0.44 [0.07; 0.82] | 0.21 [0.02; 0.40] |
| $E[Y\|X=I]$ | 0.37 [0.08; 0.66] | 0.21 [0.06; 0.35] | 0.31 [0.16; 0.46] | 0.27 [0.15; 0.39] |
| $E[Y\|X=II]$ | 0.30 [0.08; 0.53] | 0.14 [0.01; 0.27] | 0.49 [0.03; 0.95] | 0.24 [0.10; 0.38] |
| $E[Y\|X=III]$ | 0.27 [0.04; 0.51] | 0.17 [0.03; 0.32] | 0.31 [-0.01; 0.63] | 0.18 [0.07; 0.30] |
| $E[Y\|X=IV]$ | 0.26 [0.02; 0.49] | 0.21 [0.10; 0.33] | 0.17 [0.02; 0.33] | 0.19 [0.04; 0.34] |
| $\theta(\widehat{P}_n)$ | 0.33 [0.04; 0.61] | 0.35 [0.28; 0.43] | 0.37 [0.12; 0.62] | 0.29 [0.07; 0.51] |

**Summary.** Table A7 summarises the effect of our debiasing strategy on bias, SE and coverage in the simulation study. Our strategy results in uniformly valid coverage for the population mean, allowing for honest inference. For the regression coefficient, coverage was clearly improved but may remain anti-conservative for some DGMs. This may originate from residual overfitting bias inherent to these DGMs that could not be removed since (efficient) sample splitting was not performed (see Appendix A.3).

Table A7: Behaviour of estimators in debiased synthetic data in the simulation study.

| Model | Mean age | | | Effect therapy | | |
|---|---|---|---|---|---|---|
| | Bias | SE | Coverage | Bias | SE | Coverage |
| **CTGAN (SDV)** | unbiased | root-$n$ & unbiased | nominal at all $n$ | bias at small $n$ | root-$n$ & underestimated | anti-conservative at all $n$ |
| **CTGAN (Synthcity)** | unbiased | root-$n$ & unbiased | nominal at all $n$ | unbiased | root-$n$ & underestimated | anti-conservative at all $n$ |
| **TVAE (SDV)** | unbiased | root-$n$ & unbiased | nominal at all $n$ | bias at small $n$ | root-$n$ & unbiased | nominal only at large $n$ |
| **TVAE (Synthcity)** | unbiased | root-$n$ & unbiased | nominal at all $n$ | unbiased | root-$n$ & unbiased | nominal at all $n$ |

### A.7.5 Influence curve based estimation

Here, two estimators are estimated in the original and synthetic datasets: a maximum likelihood estimation (MLE)-based one, as used in traditional statistical analysis, and an efficient influence curve (EIC)-based one, as proposed in this paper and obtained after 5-fold cross-fitting during estimation of the nuisance parameters. Note that sample splitting was not used during the debiasing step. For the former, SEs are calculated via the regular expressions which discard the uncertainty associated with data-adaptive prediction during estimation, while for the latter, SEs are based on the EIC which acknowledges this uncertainty. Throughout, all estimated (model- or EIC-based) SEs are corrected with $\sqrt{1 + m/n}$ to acknowledge the sampling variability of synthetic data. This correction factor was initially proposed by Raab et al. (2016) for parametric synthetic data generators, but was found to be insufficient for synthetic data created by DGMs (Decruyenaere et al., 2024). In Section 3.3, we give the formula for the variance of our EIC-based estimator on the debiased synthetic data, which generalises this correction factor to the setting where synthetic data were generated by DGMs.

The results of our simulation study using the EIC-based estimators, as shown below in Figures A8-A11 and Table A8, remain unchanged as compared to using the MLE-based estimators, since no data-adaptive predictions (e.g., machine learning) were used during estimation of the nuisance parameters. If data-adaptive estimation were to be used, we expect the MLE-based estimators to be overly optimistic, while the EIC-based estimators could handle the additional uncertainty introduced by data-adaptive estimation.

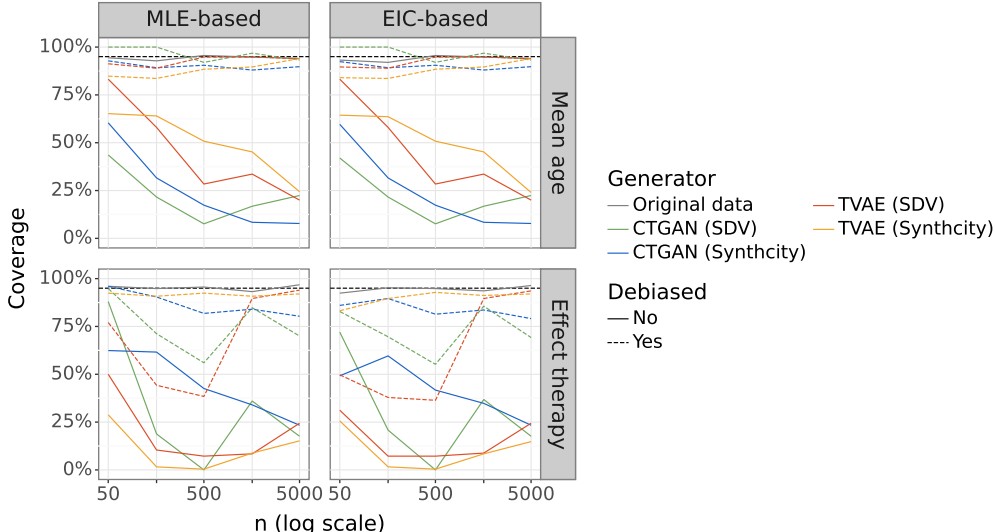

Figure A8: Empirical coverage of the 95% confidence interval for the maximum likelihood estimation (MLE)-based and efficient influence curve (EIC)-based estimators.

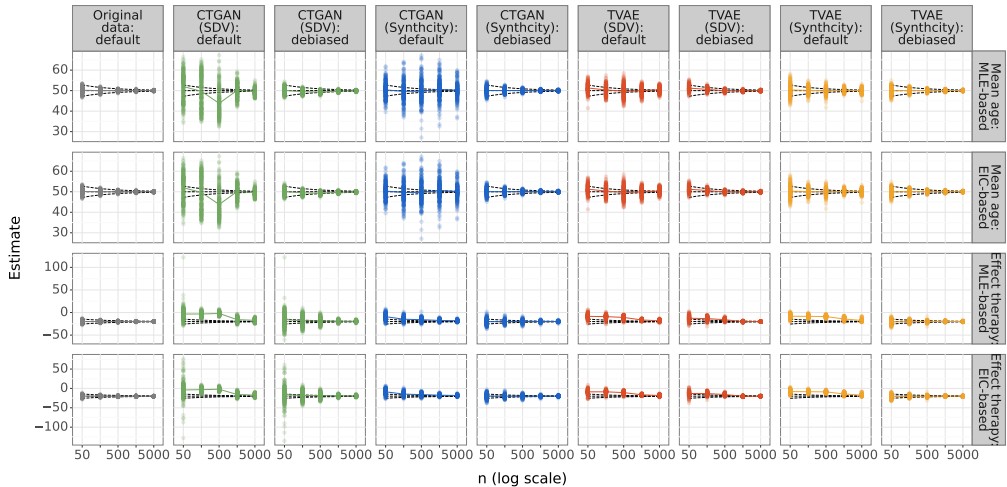

Figure A9: The horizontal dashed line represents the population parameter and each dot is a maximum likelihood estimation (MLE)-based or efficient influence curve (EIC)-based estimate per Monte Carlo run (250 dots in total per value of $n$). The dashed funnel indicates the behaviour of an unbiased and $\sqrt{n}$-consistent estimator based on observed data.

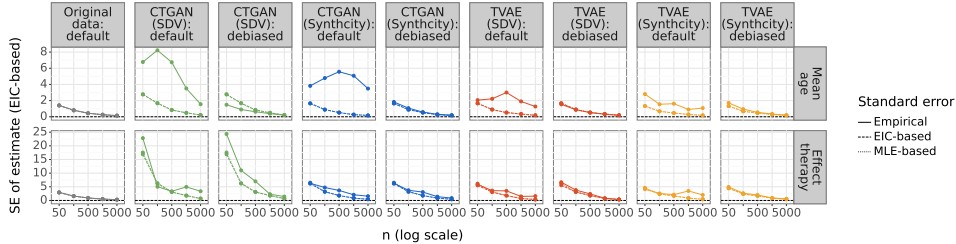

Figure A10: The empirical standard error of the efficient influence curve (EIC)-based estimators is shown. Standard errors are estimated via the maximum likelihood estimation (MLE)-based or EIC-based expressions.

Table A8: Estimated exponent $a$ [95% CI] for the power law convergence rate $n^{-a}$ for empirical SE of the maximum likelihood estimation (MLE)-based and efficient influence curve (EIC)-based estimators.

| Estimator | CTGAN (SDV) | CTGAN (Synthcity) | TVAE (SDV) | TVAE (Synthcity) |
|---|---|---|---|---|
| **Default synthetic datasets** | | | | |
| Mean age (MLE) | 0.33 [0.04; 0.62] | 0.01 [-0.16; 0.18] | 0.10 [-0.13; 0.32] | 0.21 [0.04; 0.39] |
| Mean age (EIC) | 0.33 [0.04; 0.62] | 0.01 [-0.16; 0.18] | 0.10 [-0.13; 0.32] | 0.21 [0.04; 0.39] |
| Effect therapy (MLE) | 0.23 [-0.09; 0.56] | 0.31 [0.22; 0.40] | 0.28 [0.10; 0.46] | 0.09 [-0.13; 0.31] |
| Effect therapy (EIC) | 0.33 [-0.11; 0.78] | 0.31 [0.24; 0.38] | 0.30 [0.13; 0.48] | 0.10 [-0.13; 0.32] |
| **Debiased synthetic datasets** | | | | |
| Mean age (MLE) | 0.42 [0.36; 0.48] | 0.50 [0.46; 0.54] | 0.45 [0.44; 0.47] | 0.47 [0.44; 0.50] |
| Mean age (EIC) | 0.42 [0.36; 0.48] | 0.50 [0.46; 0.54] | 0.45 [0.44; 0.47] | 0.47 [0.44; 0.50] |
| Effect therapy (MLE) | 0.58 [0.43; 0.73] | 0.43 [0.31; 0.55] | 0.61 [0.43; 0.79] | 0.46 [0.38; 0.55] |
| Effect therapy (EIC) | 0.63 [0.50; 0.75] | 0.43 [0.32; 0.54] | 0.64 [0.48; 0.79] | 0.46 [0.38; 0.55] |

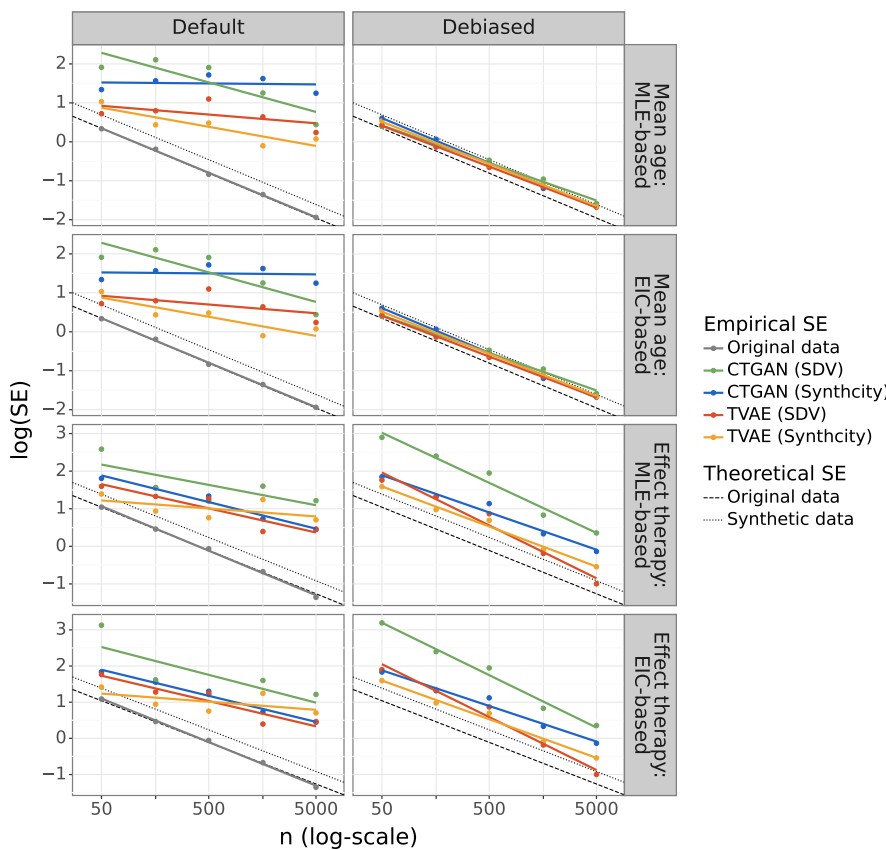

Figure A11: Convergence rate of the empirical standard error (SE) for the maximum likelihood estimation (MLE)-based and efficient influence curve (EIC)-based estimators. If the SE is of the form $SE = cn^{-a}$, where $c$ is a constant, then $\log(SE) = \log(c) + (-a)\log(n)$. Therefore slope $a$ represents the convergence rate and the vertical offset $\log(c)$ indicates the log asymptotic variance. The dashed line indicates the behaviour of the SE of an unbiased and $\sqrt{n}$-consistent estimator based on observed data, whereas the dotted line indicates the assumed behaviour of the SE of the same estimator based on synthetic data, following the correction proposed by Raab et al. (2016).

## A.8 Case studies

### A.8.1 International Stroke Trial

We adapt the framework discussed in Section 4.1 to the International Stroke Trial (IST), one of the biggest randomised trials in acute stroke research (Sandercock et al., 2011). The dataset with 19285 complete cases now constitutes our *population*. We mimic different hypothetical settings where an institution only has access to a limited *sample* of observations, with the sample size $n$ varying between 50 and 5000.

In order to easily share the data with other researchers, the institution generates a synthetic dataset with sample size $m$, where $m = n$. Similarly to the simulation study, we repeated this process 100 times per sample size $n$, to be able to calculate the empirical coverage levels. For illustration purposes, we focus on the effect of aspirin on the outcome at 6 months and report the proportion of deaths for the two treatment arms (aspirin and no aspirin), and its corresponding risk difference.

For each value of $n$, two default synthetic datasets were generated using both CTGAN and TVAE. Given the interest in the proportion of death in the group with and without aspirin, we use the debiasing strategy with respect to the population mean. This implies that the default synthetic dataset was first split by treatment and then debiased with relation to the population mean within each treatment arm. The two debiased subdatasets were then afterwards combined into one debiased synthetic dataset for each generative model. For both the default and debiased synthetic dataset, the sampling variability of synthetic data is acknowledged by inflating the standard errors (SEs) by the correction factor $\sqrt{1 + m/n}$.

The funnel plots for the proportion of deaths in both treatment arms and the risk difference are shown in Figure A12. We noticed that using the same hyperparameters as in the simulation study resulted in biased estimates, as can be seen in Figure A12. For this reason, we highlight the results obtained by training with the default hyperparameters suggested by the package SDV (Patki et al., 2016) instead. Analogously to the simulation study, our debiasing strategy decreases the variance of the mean estimator in both treatment arms, remedying the slower-than-$\sqrt{n}$-convergence observed in the default synthetic datasets. The impact for the applied researcher can be better understood by looking at the empirical coverage levels of the $95\%$ CI for the true proportion of deaths in the aspirin arm, for all sample sizes and DGMs considered. Figure A13a illustrates that in contrast to the default synthetic datasets, the coverage levels based on the debiased synthetic datasets are all positioned around the nominal level.

One of the original research questions in Sandercock et al. (2011) was whether or not there is a difference in risk of death between the treatment arms. Figure A13b depicts the empirical type 1 error rate for the risk difference in death between aspirin and no aspirin group based on original data, default and debiased synthetic data. For the aforementioned reason, we focus on the results obtained by training with the default hyperparameters suggested by the package SDV (Patki et al., 2016). Should the researcher use the default synthetic data, they would very often falsely conclude that the risk is significantly different from the true difference of $-0.009$, as calculated based on our population (the full dataset), while using the debiased synthetic dataset basically eliminates this high number of false-positives, as is the case in the original data as well.

Table A9: Estimated exponent $a$ [95% CI] for the power law convergence rate $n^{-a}$ for empirical SE. Note that a convergence rate could not be estimated for the default synthetic data when hyperparameters suggested by the package Synthcity. This occurred because there was no variance in the estimates for sample sizes of 1600 and 5000.

| Estimator | Original | CTGAN (SDV) | CTGAN (Synthcity) | TVAE (SDV) | TVAE (Synthcity) |
|---|---|---|---|---|---|
| **Default synthetic datasets** | | | | | |
| Proportion death aspirin group | 0.54 [0.50; 0.59] | 0.02 [-0.06; 0.11] | NaN [NaN; NaN] | 0.23 [-0.17; 0.62] | NaN [NaN; NaN] |
| Proportion death no aspirin group | 0.54 [0.52; 0.57] | 0.02 [-0.15; 0.18] | NaN [NaN; NaN] | 0.23 [-0.14; 0.61] | NaN [NaN; NaN] |
| Risk difference death | 0.54 [0.51; 0.56] | 0.01[-0.24; 0.26] | NaN [NaN; NaN] | 0.46 [0.14; 0.78] | NaN [NaN; NaN] |
| **Debiased synthetic datasets** | | | | | |
| Proportion death aspirin group | - | 0.54 [0.48; 0.61] | 0.59 [0.52; 0.67] | 0.53 [050; 0.56] | 0.60 [0.51; 0.70] |
| Proportion death no aspirin group | - | 0.52 [0.46; 0.59] | 0.58 [0.55; 0.62] | 0.53 [0.49; 0.58] | 0.58 [0.49; 0.67] |
| Risk difference death | - | 0.55 [0.48; 0.62 ] | 0.57 [0.53; 0.60] | 0.53 [0.50; 0.57] | 0.57 [0.53; 0.61] |

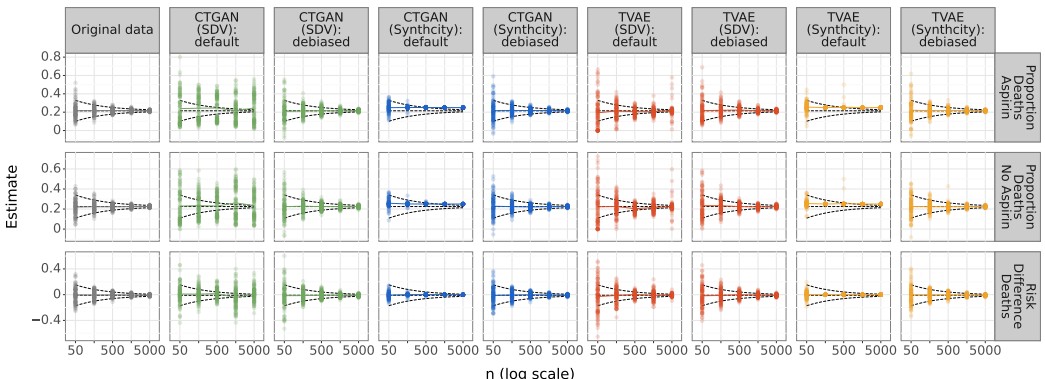

Figure A12: Estimates for the proportion of death in both treatment arms and their corresponding risk difference estimates. We show results for original data, and both default synthetic data (left) and debiased synthetic data (right) for all four generators. The horizontal dashed line represents the population proportion of death in each group and the corresponding risk difference, and each dot is an estimate per Monte Carlo run (100 dots in total per value of n). The dashed funnel indicates the behaviour of an unbiased and $\sqrt{n}$-consistent estimator based on observed data.

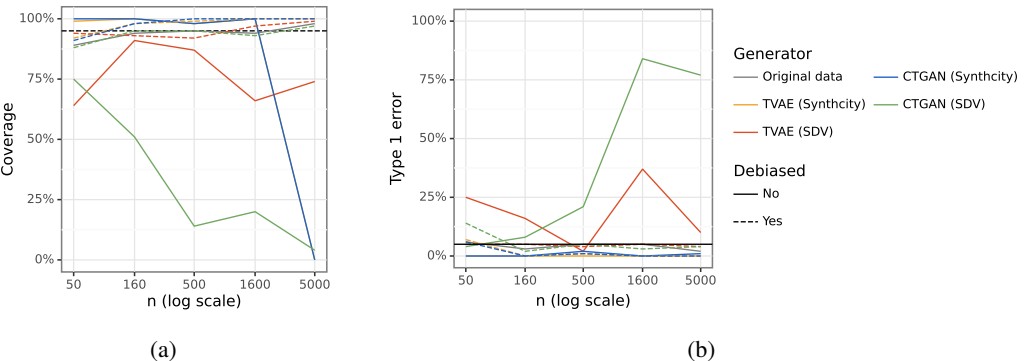

(a)                         (b)

Figure A13: Figure (a) shows the empirical coverage of the 95% CI for the true proportion of death in the aspirin treatment arm. In Figure (b), one can find the empirical type 1 error rate for the risk difference in death between aspirin and no aspirin group based on original data, default and targeted synthetic data. The null hypothesis states that the risk difference is equal to $-0.009$, the risk difference as observed in the population (i.e. the original IST data). Tests were conducted at the 5% significance level, where the black horizontal line on the figure depicts this nominal level.

