# OpenReview forum: "Debiasing Synthetic Data Generated by Deep Generative Models"
_NeurIPS.cc/2024/Conference — NeurIPS 2024 poster_

### Official Review · Reviewer_ZgH6 · 2024-07-12

**Soundness:** 3
**Presentation:** 3
**Contribution:** 3
**Rating:** 7
**Confidence:** 3

**Summary:**

The authors propose a method for debiasing DGM-generated tabular data such that the population average of the EIC for the observed data distribution and its estimation becomes zero, resulting in elimination of the corresponding biasing term. This is done through augmenting the DGM output such that the population mean of the synthetic data matches the sample average of the observed data. As such, the method is generator-agnostic. The authors validate their approach through extensive experiments, demonstrating significant reductions in bias while maintaining data utility.

**Strengths:**

1. The paper is technically sound with in-depth theoretical analysis that are backed up by practical examples. We appreciate the additional detailed investigations presented in the Appendix, which shows the dedication of the authors to do a robust evaluation of their methodology.

2. The paper tackles the critical issue of bias in synthetic data, a relatively underexplored area. The proposed method integrates fairness constraints directly into the data generation process, a novel approach, as opposed to post-hoc fairness adjustments. The proposed methodology specifically addresses the bias term concerning the observed data distribution, O, and its DGM estimation, P, as opposed to previous efforts that (are said to) have focused on the bias term describing the discrepancy between the sampled estimated (synthetic) distribution, S, and the approximation of P as P¨ (P-hat) through S.

3. The paper is very well written and well organized (it was a pleasure to read). Figures and tables are well-presented and support the narrative effectively.

4. The results have significant implications for improving fairness in machine learning, an area of growing importance. The proposed method for debiasing synthetic tabular data is straightforward and well described and hence easily implemented and adaptable by others

**Weaknesses:**

1. The paper would benefit from a more in-depth evaluation of potential limitations or edge cases where the method might not perform as well.

2. It is difficult to assess the novelty in the proposed method as there is no presentation of related works nor comparison with alternative methods. A more detailed review of related work would help contextualize the novelty of the proposed method.

3. It would be good to put the work in contrast with related works would improve the paper and elucidate the authors contribution to the field.

4. The significance is still difficult to assess due to the aforementioned lack of comparison to alternative approaches. Additionally, potential impacts on different domains (e.g., healthcare, finance) could be explored to highlight the broader significance.

**Questions:**

1. Could the authors provide more details on how their approach scales with larger datasets and more complex models?
2. How does the proposed method perform in scenarios with multiple sources of bias or more intricate data dependencies?
3. Can the authors elaborate on any potential trade-offs between debiasing effectiveness and computational efficiency?
4. How is "a very large sample" defined? L146 says 1,000,000 observations is "large", but the quality analysis in Section A.7.2 has max n = 5,000.
5. Is the necessary synthetic data sample size generalizable or does it depend on the complexity of the parameter space?

**Limitations:**

The authors properly address limitations of their work. However, they could further discuss the scalability of the proposed method to large datasets.

---

> ### Author Rebuttal · Authors · 2024-08-07
>
> We would like to thank Reviewer ZgH6 for their careful reading of the manuscript and constructive review.
>
> **It would be good to put the work in contrast with related works <...>.**
>
> We noted that other reviewers raised similar questions and have therefore included an extended comparison with related work in the global rebuttal, which will also be integrated into the manuscript.
>
> **Could the authors provide more details on how their approach scales with larger datasets and more complex models?**
>
> We elaborate on this in our global response as well, but agree that future work should additionally investigate high-dimensional settings. Complex joint distributions of the observed data may impede the baseline convergence of the DGM, which in turn could diminish the utility gain provided by our debiasing approach. This is a limited weakness of our proposal as it may be the unavoidable consequence of working with high-dimensional data (e.g., debiased machine learning strategies likewise provide no guarantees in high-dimensional complex settings). Nevertheless, we expect that our approach will still improve inference, as is likewise observed for debiased estimators based on machine learning predictions.
>
> Concerning an extension to more complex estimators, we would like to highlight that the proposed debiasing strategy is already applicable to all pathwise differentiable parameters. This includes essentially all finite-dimensional parameters used in routine practice, such as variances, regression coefficients,... An empirical study for other parameters than the population mean and linear regression coefficient is of interest, but beyond the scope of this work, whose aim is to show, for the first time, that valid inference on DGM-based synthetic data is feasible, but requires proper debiasing. Lastly, we would like to stress that our proposed strategy is generator-agnostic and can therefore be extended to more complex DGMs.
>
> **How does the proposed method perform in scenarios with multiple sources of bias or more intricate data dependencies?**
>
> Regarding the first part of the question concerning the multiple sources of bias, we wonder whether there is a misunderstanding regarding bias in the context of fairness. In view of this, we clarified the meaning of bias in our manuscript in the global rebuttal. In particular, we focus on the estimation bias that arises in estimators based on synthetic data generated by DGMs. In our paper, we
> target each estimator separately (e.g. the mean of age vs. the effect of therapy on blood pressure), but future work should focus on targeting all these sources of bias at once. This entails that the person generating the synthetic data is aware of all the analyses that will be run on those data, and has access to the corresponding EICs. Whether targeting multiple estimators at once affects the efficacy of the debiasing approach remains to be studied. The effect of more intricate data dependencies on the performance of our debiased strategy is an interesting yet open research question at this point. Our approach relies on fast baseline convergence of the DGM, which is influenced by the dimension of the data and the complexity of the true data-generating model.
>
> **Can the authors elaborate on any potential trade-offs between debiasing effectiveness and computational efficiency?**
>
> There are indeed potential trade-offs that should be taken into consideration. As alluded to in the paper, the current implementation of our strategy does not use sample splitting to estimate the bias term and therefore does not require the generative model to be retrained, making our strategy computationally efficient. Future extensions that necessitate a sample splitting procedure during which the generative model is retrained on different subsets of the data (and as such, computation time will depend on the dimensionality of these subsets) should focus on both computational and statistical efficiency. Our theory currently discards sample splitting due to preliminary results suggesting that the bias reduction did not outweigh the increase in finite-sample bias that may result from training the DGM on smaller sample sizes. This suggests a trade-off between debiasing effectiveness and statistical properties such as finite-sample bias and increased variability when using sample splitting.
>
> **How is “a very large sample” defined? <...> Is the necessary synthetic data sample size generalizable <...>?**
>
> In the set-up of our simulation study (which Section A.7.2 relates to), we consider different sample sizes of observed and synthetic data: $n \in$ {50, 160, 500, 1600, 5000}. These sample sizes were chosen to reflect a range that is common in real medical settings. Next, in the theory of our debiasing strategy for a linear regression coefficient we need the terms $E_{\hat{P_n}}(A|X_i)$ and $E_{\hat{P_n}}(Y|X_i)$. Given that these reflect unknown conditional population means, we need a way to approximate these in practice. In this case, one often creates an artificial “population” by generating a very large sample of e.g. one million observations, to then approximate the population mean by calculating the sample mean of this Monte Carlo sample. The purpose of this very large sample is merely to recreate a population of samples generated by the trained DGM and is thus not comparable to the magnitude of sample sizes of the observed data. We believe one million observations will suffice to capture the distribution of samples generated by the DGM, regardless of the specific type of DGM.
>
> Chernozhukov, V., Chetverikov, D. et al. (2018). Double/debiased machine learning for treatment and structural parameters. The Econometrics Journal, 21(1):C1–C68.
>
> Decruyenaere, A., Dehaene, H. et al. (2024). The real deal behind the artificial appeal: Inferential utility of tabular synthetic data. In The 40th Conference on UAI.
>
> van der Laan, M. J. and Rose, S. (2011). Targeted Learning. Springer Series in Statistics.

---

> > ### Comment · Reviewer_ZgH6 · 2024-08-11
> >
> > Thanks for these clarifications. Given this, I will retain my original score.

---

### Official Review · Reviewer_4cJP · 2024-07-14

**Soundness:** 3
**Presentation:** 3
**Contribution:** 3
**Rating:** 7
**Confidence:** 1

**Summary:**

This paper addresses the significant biases introduced in synthetic data generated by deep generative models (DGMs) that compromise the inferential utility of such data. The authors propose a new debiasing strategy based on techniques adapted from debiased or targeted machine learning. Their approach aims to reduce biases and enhance convergence rates, thereby improving the reliability of statistical analyses performed on synthetic data. The effectiveness of the proposed strategy is demonstrated through a simulation study and two case studies using real-world data.

**Strengths:**

- **Originality:** The paper tackles the important problem of bias in synthetic data generated by DGMs, which has significant implications for privacy protection and data analysis. The authors propose a novel debiasing strategy that adapts techniques from targeted machine learning to the context of synthetic data generation. This represents a new direction in addressing the challenges of inferential utility in synthetic data. The approach is generator-agnostic, making it widely applicable to various types of DGMs.
- **Quality:** The submission is technically sound, with clear explanations of the theoretical foundations and derivations of the debiasing strategy. The authors provide a simulation study and two case studies that demonstrate the effectiveness of their approach in improving the coverage of confidence intervals and enabling more reliable statistical inference from synthetic data. The paper is well-written and organized, making it easy to follow the authors' reasoning and understand their contributions.
- **Significance:** The results presented in the paper are relevant for the field of synthetic data generation and privacy protection. The proposed debiasing strategy has the potential to improve the reliability and applicability of synthetic data in statistical inference, which can have broad implications for various domains where privacy is a concern.

**Weaknesses:**

- **Limited Scope:** The simulation study and case studies focus on low-dimensional settings and a limited number of estimators (population mean and linear regression coefficient). While the positive results are encouraging, further evaluation is needed to assess the effectiveness of the debiasing strategy in high-dimensional settings and for a wider range of statistical analyses.
- **Dependence on EICs:** The proposed debiasing strategy requires access to the Efficient Influence Curves (EICs) of the target parameters of interest. This limits the applicability of the method to parameters that are pathwise differentiable and have known EICs.
- **Conditional Sampling Limitation:** The debiasing procedure for the linear regression coefficient requires sampling synthetic data conditional on covariates, which may not be feasible for all types of DGMs.

**Questions:**

- **High-Dimensional Settings:** How well does the debiasing strategy perform in high-dimensional settings where DGMs are more commonly used? Are there any challenges or limitations in applying the method to high-dimensional data?
- **Complex Estimators:** Can the debiasing strategy be extended to more complex estimators beyond the population mean and linear regression coefficient?
- **Alternative Debiasing Approaches:** Are there alternative debiasing approaches that could be explored to address the limitations of the proposed method, such as its dependence on EICs and the conditional sampling requirement?
- **Privacy Considerations:** While the paper mentions the potential increase in privacy disclosure risk with larger synthetic datasets, it would be beneficial to discuss this trade-off in more detail. How can the debiasing strategy be balanced with privacy concerns, especially in the context of differentially private synthetic data generation?
- **Failure of Assumptions**: Are there specific scenarios where the assumptions required for the debiasing method might not hold, and how would this impact the results?

**Limitations:**

The authors themselves are highly aware and forthcoming about the limitations of this study, which in my eyes is a big bonus.
- The authors discuss the low-dimensional setting of their simulations and acknowledge that DGMs might be less suited for such settings.
- They note the need for the person generating synthetic data to be aware of the analyses that will be run, which may limit the method's applicability.
- The authors note the need to condition on data covariates to generate appropriate synthetic data.
- The potential trade-off between synthetic data sample size and privacy risks is mentioned, though it is beyond the scope of this paper.

---

> ### Author Rebuttal · Authors · 2024-08-07
>
> We would like to thank Reviewer 4cJP for their careful reading of the manuscript and constructive review.
>
> **How well does the debiasing strategy perform in high-dimensional settings <...>?**
>
> We elaborate on this in our global response as well, but agree that future work should additionally investigate high-dimensional settings. Complex joint distributions of the observed data may impede the baseline convergence of the DGM, which in turn could diminish the utility gain provided by our debiasing approach. This is a limited weakness of our proposal as it may be the unavoidable consequence of working with high-dimensional data (e.g., debiased machine learning strategies likewise provide no guarantees in high-dimensional complex settings). Nevertheless, we expect that our approach will still improve inference, as is likewise observed for debiased estimators based on machine learning predictions.
>
> **Can the debiasing strategy be extended to more complex estimators <...>?**
>
> We elaborate on this in our global response, but note that the proposed debiasing strategy is already applicable to all pathwise differentiable parameters. This includes essentially all finite-dimensional parameters used in routine practice, such as variances, regression coefficients,... An empirical study for other parameters than the population mean and linear regression coefficient is of interest, but beyond the scope of this work, whose aim is to show, for the first time, that valid inference on DGM-based synthetic data is feasible, but requires proper debiasing.
>
> **Are there alternative debiasing approaches that could be explored to address the limitations of the proposed method <...>?**
>
> We refer the reviewer to our global response on related work and alternative debiasing approaches. Regarding their specific questions on the dependence on EICs and conditional sampling, future work could:
> - integrate insights from the literature on automatic debiased machine learning (Chernozhukov et al. (2022)) that allows to estimate the EIC automatically from data.
> - combine the findings from our work and work of Ghalebikesabi et al. (2022) on importance weighting, where the weights could be targeted to eliminate the impact of the data-adaptive estimation of the weights. This could relax the fast baseline convergence assumption, and enable the same ‘debiased’ synthetic data to be used for multiple analyses. The latter might also be possible by undersmoothing the DGM, by extending the recent work on kernel debiased plug-in estimation to the synthetic data context.
> - incorporate strategies for conditional sampling from DGMs (e.g. the work of Zhou et al. (2023)). Note that in case of a limited number of discrete covariates, conditional sampling can be approximated by (unconditionally) generating a large synthetic sample and taking a conditional subset, as done in our simulation study and second case study.
>
> **How can the debiasing strategy be balanced with privacy concerns <...>?**
>
> The privacy-utility trade-off is well-known in synthetic data literature and relates to the sample size of synthetic datasets: as m → +∞, the synthetic dataset has higher probability to contain synthetic records that are close to the original, resulting in higher disclosure risk (Reiter, J. and Drechsler, J. (2010)), while the inferential utility improves in terms of more precise estimates (as can be seen from the expressions for the standard error). Although formal privacy assessment is beyond the scope of our paper, this trade-off is relevant to our work. Default synthetic data created by differentially private (DP) generative models will by definition not have worse privacy with increasing sample size, but will only have better inferential utility when the estimators remain roughly √n-consistent. With DP DGMs, this is not the case (see Decruyenaere et al. (2024)), but it could possibly be attained by extending our debiasing approach, without compromising any privacy guarantees due to the post-processing immunity of DP. This extension should incorporate the additional DP-constraints into the derivation of the difference between $\theta(\tilde{P}_m)$ and $\theta(P)$. It is not clear yet what this would look like, given that different DP generators add noise in different ways (e.g. DPGAN adds noise to the gradients, while PATE-GAN adds noise to the majority vote in the training procedure). We will discuss this nuance more clearly in our revised manuscript.
>
> **Are there specific scenarios where the assumptions required for the debiasing method might not hold <...>?**
>
> Our approach relies on fast baseline convergence of the DGM: faster than n-to-the-quarter convergence rates for the unknown functionals of $\tilde{P}_m$ and $\hat{P}_n$ that appear in the EIC are required. Their attainability depends on the number of parameters in the DGM itself, the dimension of the data and the complexity of the observed data distribution. If these rates are not attained, then we expect that our approach will still improve coverage, but that no root-n consistent estimators will be obtained based on the synthetic data, resulting in increasingly anti-conservative coverage with larger sample size n.
>
> Chernozhukov, V., Newey, W., and Singh, R. (2022). Automatic debiased machine learning of causal and structural effects. Econometrica, 90(3), 967-1027.
>
> Decruyenaere, A., Dehaene, H. et al. (2024). The real deal behind the artificial appeal: Inferential utility of tabular synthetic data. In The 40th Conference on UAI.
>
> Ghalebikesabi, S., Wilde, H. et al. (2022). Mitigating statistical bias within differentially private synthetic data. In The 38th Conference on UAI.
>
> Reiter, J. and Drechsler, J. (2010). Releasing multiply-imputed synthetic data generated in two stages to protect confidentiality. Statistica Sinica, 20, 405–421.
>
> Zhou, X., Jiao, Y. et al. (2023). A deep generative approach to conditional sampling. Journal of the American Statistical Association, 118(543), 1837-1848.

---

> > ### Comment · Reviewer_4cJP · 2024-08-13
> > **Response**
> >
> > I thank the authors for their detailed and thorough response. I will keep my original score.

---

### Official Review · Reviewer_LYaP · 2024-07-15

**Soundness:** 2
**Presentation:** 1
**Contribution:** 2
**Rating:** 4
**Confidence:** 1

**Summary:**

This paper tackle the problem of creating/imputing unbiased synthetic data, and analyse the potential bias brought by these imputation methods.

**Strengths:**

The tackled problem is important and very timely, and linked to the general question of biais of generative models [1].


[1] Wyllie, Sierra, Ilia Shumailov, and Nicolas Papernot. "Fairness feedback loops: training on synthetic data amplifies bias." The 2024 ACM Conference on Fairness, Accountability, and Transparency. 2024.

**Weaknesses:**

I do not know if this comes from my lack of knowledge in the field, but I found the paper very hard to understand, no clear theoretical result is encapsulated, and it is not even very clear to me what is proposed (I guess this is what is after line 91).
I thus have a lot of questions:
- line 90, how exactly $S_i$ is sampled? Are you sampling first $A_i$ and $X_i$, and then $Y_i|A_i, X_i$?
- What is your main theoretical result? Is it Equation 2? or Section 3.3?
- What is $\phi$ (I am not sure it is defined in the main text)
- How is the deep generative model learned in the first place?

**Questions:**

see above

**Limitations:**

yes

---

> ### Author Rebuttal · Authors · 2024-08-07
>
> We would like to thank Reviewer LYaP for their careful reading of the manuscript and constructive review.
>
> **I do not know if this comes from my lack of knowledge in the field, but I found the paper very hard to understand.**
>
> We are aware that the proposed strategy is not trivial and combines different concepts in order to eliminate the deviating behaviour of estimators seen in synthetic data (Decruyenaere et al., 2024). However, we hope that the proposed guidance below for Section 3 of a
> revised manuscript improves readability.
> We will also rewrite the introduction to stress the difference with other related work and emphasize our contribution in the field. Therefore, we have included an extended comparison with related work in the global rebuttal. We summarize our contributions as follows.
> Inferential utility of synthetic data is compromised when data are generated with deep generative models (DGMs), even when using earlier proposed correction factors (Decruyenaere et al., 2024). In other related work, procedures were developed relying on multiple synthetic datasets or focussing on only statistical generators (and hence implicitly rely on root-_n_ consistent estimators). In this manuscript, we propose a debiasing strategy that allows to create synthetic data via DGMs while maintaining valid inference for a population parameter.
>
> **Line 90, how exactly $S_i$ is sampled? Are you sampling first $A_i$ and $X_i$, and then $Y_i | A_i, X_i$?**
>
> In this work, we focus on DGMs but within this setup, the proposed strategy is generator-agnostic. Therefore, the specific sampling procedure depends on the generator that is used in specific cases. In our simulation and case studies, we apply TVAE and CTGAN (Xu et al., 2019) to generate synthetic samples. In these algorithms, the trained decoder (i.e. TVAE) or generator (i.e. CTGAN) are used to generate synthetic samples where all features are sampled jointly as $Y_i, A_i, X_i$ (and thus not conditional as $Y_i | A_i, X_i$).
>
> In our proposed method, we perform a post-processing of the obtained synthetic samples to eliminate the bias that would otherwise occur when the parameter of interest is estimated based on synthetic samples. The exact procedure for this post-processing is explained in Section 3.1 and 3.2. In Section 3.2 it can be seen that we need to estimate e.g. $E_{\widehat{P}_n}(Y|X_i)$ and this requires the generation of synthetic values of $Y$ conditional on a given observed level $X_i$. This can be circumvented by generating, based on the DGM, a very large synthetic sample, taking the subset where $X$ equals $X_i$, and estimating the mean within this subset. If $X$ were continuous, then the DGM would necessitate a built-in conditional sampling module, which is not a given in any type of DGM, and lies beyond the scope of our simulation study.
>
> **What is your main theoretical result? Is it Equation 2?**
>
> Our main theoretical result can be found in Sections 3.1 and 3.2, where we propose a debiasing strategy for two estimators: the population mean and the linear regression coefficient. In short, we propose to shift the DGM-based synthetic data.
> The remainder of Section 3 provides the theoretical background for why this debiasing approach works. We understand the confusion given the detailed yet dense explanation within the page limit. In order to prevent future confusion, we will slightly rewrite this part to ensure that the setup of Section 3 is clear. In the remainder, we provide an outline of how we would guide the reader through Section 3.
>
> We first aim at establishing an understanding of the deviation when the parameter of interest is estimated on synthetic versus observed data (i.e. $\theta(\widetilde{P}_m)-\theta(P)$). By doing so, the equation provides us with eight elements that we then further explore. We start with the two remainder terms and the two empirical process terms and elaborate on why we foresee that these terms are negligible under certain conditions. In the next paragraph, we state that we foresee no problems with terms (3) and (4), but that terms (5) and (6) however will cause bias if not taken into account.
> We then proceed by suggesting a new approach to tackle those two latter terms. For term (5) we rely on theory proposed by van der Laan and Rose (2011) and Chernozhukov et al. (2018) by analyzing the data with debiased estimators.
>
> Our contribution lies in the way we handle term (6). Finding a solution for this bias term relies on the efficient influence curve and therefore on the parameter of interest. In the next two subsections, we specifically worked out the proposed strategy for the population mean and a linear regression coefficient and we pinpoint how bias term (5) and (6) can be resolved. We finalize the Methodology section with stating the properties of the resulting estimator (i.e. the debiased estimator based on synthetic data).
>
> **What is $\phi$?**
>
> $\phi(\cdot)$ is defined as the efficient influence curve on line 75, but it could enhance readability to repeat this on line 98-99.
>
> **How is the deep generative model learned in the first place?**
>
> We are not sure that we fully understand this question as we rely on commonly used generators such as TVAE and CTGAN (Xu et al., 2019) to generate synthetic data. They are trained on original data and our proposed strategy does not alter this standard training phase.
>
> Chernozhukov, V., Chetverikov, D. et al. (2018). Double/debiased machine learning for treatment and structural parameters. _The Econometrics Journal_, 21(1):C1–C68.
>
> Decruyenaere, A., Dehaene, H. et al. (2024). The real deal behind the artificial appeal: Inferential utility of tabular synthetic data. In _The 40th Conference on UAI._
>
> van der Laan, M. J. and Rose, S. (2011). _Targeted Learning._ Springer Series in Statistics. Springer New York, New York, NY.
>
> Xu, L., Skoularidou, M. et al. (2019). Modeling tabular data using conditional GAN. _Advances in neural information processing systems_, 32.

---

### Official Review · Reviewer_eYfd · 2024-07-21

**Soundness:** 3
**Presentation:** 3
**Contribution:** 2
**Rating:** 5
**Confidence:** 4

**Summary:**

The paper considers the problem of debiasing synthetic data which can have signficant issues when used for statistical analysis. In particular, they show examples of mean estimation of a variable and parameter estimate of a regression model and demonstrate the benefits of the post-processing step of their approach (model agnostic).

Overall: The paper is clearly written and the problem is well-defined. The approach is for debiasing is simple but effective and experiments validate the approach. Unclear, how it can work for more real-world settings.

**Strengths:**

(i) The problem of bias in the synthetic data is well explained by considering the two examples of population mean and the coefficient estimator in linear regression. The debiased estimators are derived for these two settings.

(ii) Experiments are first shown the synthetic datasets and the debiased estimators have very good coverage as opposed to without debiasing. The convergence rate with O(1/sqrt(n)) is al
so shown when debiased. Two real-world datasets are also considered and the issue of obtaining false conclusions from the synthetic data vs the debiased setting is illustrated.

**Weaknesses:**

(a) As mentioned in the discussion settings, the datasets are low-dimensional and the estimators are very simple such as mean and parameter estimate for linear regression.

(b) It is unclear how one can generalize these to settings where the estimators are much more complex/ubiquitous such as the covariance matrix or group fairness.

**Questions:**

(1) The references are a bit sparse on the synthetic data literature and would it be possible to enhance it with appropriate hooks? There are quite a few surveys available now.
(2) While the results are interesting, it is unclear how much is innovative in this work as opposed to building on the results from Chernozhukov, Decruyenaere and other cited works. Can you elucidate that clearly?
(3) Also, what would be needed to handle more interesting estimators for downstream applications?

---

> ### Author Rebuttal · Authors · 2024-08-07
>
> We would like to thank Reviewer eYfd for their careful reading of the manuscript and constructive review.
>
> **What would be needed to handle more interesting estimators for downstream applications? It is unclear how one can generalize these to settings where the estimators are much more complex/ubiquitous such as the covariance matrix or group fairness.**
>
> Given the importance of this concern, we expand on this topic in the global response to all reviewers. We share the concern that theory should be aligned with practical real-world settings and we advocate that future research should focus on extending our results to broader settings. However, Decruyenaere et al. (2024) have shown that inferential utility is compromised when statistical analysis is performed on synthetic data created by deep generative models (DGMs), even in low-dimensional settings.
> We see addressing these low-dimensional settings as a necessary first step, before tackling more complex settings where valid inference is already challenging in observed data, let alone synthetic data.
>
> **The references are a bit sparse on the synthetic data literature and would it be possible to enhance it with appropriate hooks? There are quite a few surveys available now.**
>
> Indeed, it would be useful to contrast our work more extensively with additional recent research on synthetic data in the context of inferential utility.
> Other reviewers raised the same question and therefore we elaborate on this in the global rebuttal. There, we provide a comparison of our work with related literature, which we will add to the revised manuscript.
> We are convinced that this addition will improve our manuscript and will better capture how our work stands out w.r.t. other contributions.
>
> **While the results are interesting, it is unclear how much is innovative in this work as opposed to building on the results from Chernozhukov, Decruyenaere and other cited works. Can you elucidate that clearly?**
>
> For a global overview of related work and our specific contribution, we would like to refer to the global rebuttal. In addition, we will address the specific question related to the work of Chernozhukov et al. (2018) and Decruyenaere et al. (2024).
>
> Our debiasing strategy originates from the observation in Decruyenaere et al. (2024) that inferential utility is compromised when naive statistical analyses are performed on synthetic data created by deep generative models (DGMs), even when applying a previously proposed correction factor to the standard error.
> They claim that this can be attributed in great extent to the additional layer of uncertainty resulting from the synthetic data generation process and the deviating convergence rate of parameters when estimated in synthetic data, leading to unreliable confidence intervals and overly confident conclusions made from these synthetic data. In our work, we first aim to establish an understanding of this deviation when the parameter of interest is estimated based on a synthetic versus observed sample (i.e. $\theta(\widetilde{P}_m)-\theta(P)$). By doing so, we are able to identify two bias terms which can be targeted in order to decrease this deviation between synthetic and original data.
>
> For the first bias term (referred to as bias term (5) in the manuscript), we rely on theory proposed by van der Laan and Rose (2011) and Chernozhukov et al. (2018), allowing us to analyze the data with debiased estimators.
> Our contribution lies in the way we handle the second bias term (i.e. bias term (6) in the manuscript). As stated in the manuscript, finding a solution for this bias term relies on the efficient influence curve and therefore on the target parameter of interest. In the first two subsections of the Methodology section, we specifically work out a strategy for the population mean and the linear regression coefficient, showing how our debiasing strategy combines a solution for both bias term (5) and (6).
> We finalize the Methodology section with stating the properties of the resulting estimator (i.e. the debiased estimator based on the synthetic data).
>
> To summarize, we show how the problems concerning inferential utility can be removed, by targeting synthetic data generated by DGMs to perform optimally for the specific data analysis that is envisaged. This builds on ideas from debiased or targeted machine learning (van der Laan and Rose (2011), Chernozhukov et al. (2018)), but is nonetheless a non-trivial extension as (a) those works do not consider synthetic data; and (b) it required us to work out how the estimation errors in the DGM propagate into the estimator calculated on synthetic data, and based on this, how those data must be adapted to dampen propagation of those errors.
> As far as we are aware, our approach is the only one that provides formal guarantees for (more) reliable inference from synthetic data created by deep generative models.
>
>
> Chernozhukov, V., Chetverikov, D. et al. (2018). Double/debiased machine learning for treatment and structural parameters. _The Econometrics Journal_, 21(1):C1–C68.
>
> Decruyenaere, A., Dehaene, H. et al. (2024). The real deal behind the artificial appeal: Inferential utility of tabular synthetic data. In _The 40th Conference on Uncertainty in Artificial Intelligence._
>
> van der Laan, M. J. and Rose, S. (2011). _Targeted Learning._ Springer Series in Statistics. Springer New York, New York, NY.

---

> > ### Comment · Reviewer_eYfd · 2024-08-14
> >
> > Thanks to the authors for the detailed rebuttal. I have updated my score to 5 based on all the comments and I think it needs some work to place it appropriately in the literature.

---

### Author Rebuttal · Authors · 2024-08-07

We would like to thank all reviewers for their careful reading of the manuscript and the constructive reviews. We were pleased to read that the presentation of our work was positively received by **Reviewers eYfd, 4cJP** and **ZgH6**, though we agree with **Reviewer LYaP** that the methodology section could benefit from further clarification.
We noticed that there were three recurring topics, which we have chosen to address in this global response, while elaborating further in reviewer-specific rebuttals where needed.

**Reviewer eYfd, 4cJP** and **ZgH6** raised concerns about the generalizability of our results to high-dimensional settings and more complex estimators. We agree with the reviewers' point. However, we would like to highlight that our paper is, to the best of our knowledge, the first to demonstrate, with significant generality, how valid statistical inferences can be derived from synthetic data. This addresses a critical yet largely overlooked issue with synthetic data: standard analyses of synthetic data based on deep generative models (DGMs) often result in significantly biased inferences, leading to confidence intervals that rarely contain the true parameter values.
Decruyenaere et al. (2024) have shown that this problem persists even for basic statistics such as means and regression coefficients, which are central to most routine analyses. This is why our current paper initially focuses on proposing a solution for these fundamental quantities. Moreover, our methodology is applicable to all pathwise differentiable parameters, encompassing essentially all finite-dimensional parameters, including variances, correlations, average treatment effects, and regression coefficients in common statistical models.

While empirical investigation of our proposed solution across these various contexts is undoubtedly important, it falls beyond the scope of the present work.

We noted that **Reviewer LYaP** and **ZgH6** linked our work to the field of fairness in machine learning. We want to make clear that when we talk about bias, we mean statistical bias, being the average difference between the estimate obtained in the data and the true underlying parameter. We do not mean to remove bias towards some particular subpopulation, which is how this term is often used in fairness literature. We apologize if this caused confusion. We do agree that fairness is an important concept, and future work could therefore focus on extending our debiasing approach to metrics like group fairness, as part of the extensions discussed in our previous point.

Finally, most reviewers suggest to clarify whether alternative approaches exist to obtain valid inference from synthetic data and to contrast these with our approach.
While several approaches have been proposed to account for the uncertainty arising from synthetic data generation, we are not aware of strategies for generating and/or analyzing DGM-based synthetic data that guarantee valid inference.
Raghunathan et al. (2003) developed a framework inspired on the work of multiple imputation for missing data, by combining the results of multiple synthetic datasets, but this is not readily applicable to DGM-based synthetic data.
Räisä et al. (2023a) extended this work for differentially private (DP) synthetic data, acknowledging the additional DP noise during synthetic data generation, but continue to consider parametric (Bayesian) data-generation strategies.

Our work instead focuses on obtaining valid inference from a single synthetic dataset, which is more attractive for use by practitioners.
The method suggested by Awan and Cai (2020) to preserve efficient estimators in a single (DP or non-DP) synthetic dataset relies on generating synthetic data conditional on the estimate in the original data and is only applicable to parametric generative models and therefore suffers the same limitation as the aforementioned approaches.
To allow for Bayesian inference from a single DP synthetic dataset, Wilde et al. (2021) proposed a corrected analysis that relies on the availability of additional public data, while Ghalebikesabi et al. (2022) investigated importance weighting methods to remove the noise-related bias, but they do not study the impact on inference.
Relative to the aforementioned approaches, our work is the first to consider the impact of the typical slower-than-$\sqrt{n}$-convergence of estimators in (DP or non-DP) synthetic data created by deep generative models. As far as we are aware, our approach is thus the only one that provides some formal guarantees for (more) honest inference in this setting. We will add this paragraph on related work to our revised manuscript.

Awan, J., and Cai, Z (2020). One Step to Efficient Synthetic Data. _arXiv preprint arXiv_:2006.0239.

Decruyenaere, A., Dehaene, H. et al. (2024). The real deal behind the artificial appeal: Inferential utility of tabular synthetic data. In _The 40th Conference on Uncertainty in Artificial Intelligence._

Ghalebikesabi, S., Wilde, H. et al. (2022). Mitigating statistical bias within differentially private synthetic data. _Proceedings of the Thirty-Eighth Conference on UAI_, in _Proceedings of Machine Learning Research_ 180:696-705.

Raghunathan, T. E., Reiter, J. P., and Rubin, D. B. (2003). Multiple imputation for statistical disclosure limitation. _Journal of official statistics_, 19(1), 1.

Räisä, O., Jälkö, J. et al. (2023a). Noise-aware statistical inference with differentially private synthetic data. _Proceedings of The 26th International Conference on AISTATS_, in _Proceedings of Machine Learning Research_, 206, 3620–3643.

Räisä, O., Jälkö, J., and Honkela, A. (2023b). On Consistent Bayesian Inference from Synthetic Data. _arXiv preprint arXiv_:2305.16795.

Wilde, H., Jewson, J. et al. (2021). Foundations of Bayesian learning from synthetic data. _Proceedings of The 24th International Conference on AISTATS_, in _Proceedings of Machine Learning Research_, 130, 541–549.

---

### Decision · Program_Chairs · 2024-09-25

**Decision:**

Accept (poster)

**Comment:**

This paper proposes a debiasing strategy for synthetic data generated by deep generative models (DGMs), which aims to eliminate the bias that arises when estimating population parameters from synthetic data. The authors argue that existing methods for generating synthetic data can lead to biased estimates, and propose a new approach that involves post-processing the synthetic data to remove the bias. The method is generator-agnostic, meaning it can be applied to different types of DGMs, and is based on techniques adapted from debiased or targeted machine learning.

The reviewers praised the paper for its technical soundness, originality, and significance, noting that it addresses an important problem in the field of synthetic data generation. However, they also raised some concerns about the paper's readability and clarity, particularly for readers without a strong background in the field. The reviewers also asked for more information about the scalability of the proposed method to larger datasets and more complex models, and about the potential trade-offs between debiasing effectiveness and computational efficiency.

The authors responded to these concerns in their rebuttal, providing additional explanations and clarifications, and acknowledging the limitations of their study. They also suggested avenues for future research, including investigating the performance of the method in scenarios with multiple sources of bias or more intricate data dependencies.

After the rebuttal and discussion, three reviewers lean towards acceptance, while one leans towards rejection. However, the latter reviewer reports only a confidence of 1 and has not acknowledged the authors' rebuttal. Overall, I would thus recommend to accept the submission. However, I would further recommend that the authors seriously take the reviewer feedback into account for the camera-ready version, especially when it comes to the clarity of the presentation of the method and theoretical results to a wider ML audience and the positioning of this work within the related literature.